# Towards Faster Decentralized Stochastic Optimization with Communication Compression

**Rustem Islamov**[1*] **Yuan Gao**[2,3*] **Sebastian U. Stich**[3]
[1]Universität Basel, [2]Universität des Saarlandes, [3]CISPA Helmholtz Center for Information Security
`rustem.islamov@unibas.ch`, `{yuan.gao, stich}@cispa.de`

## Abstract

Communication efficiency has garnered significant attention as it is considered the main bottleneck for large-scale decentralized Machine Learning applications in distributed and federated settings. In this regime, clients are restricted to transmitting small amounts of compressed information to their neighbors over a communication graph. Numerous endeavors have been made to address this challenging problem by developing algorithms with compressed communication for decentralized non-convex optimization problems. Despite considerable efforts, current theoretical understandings of the problem are still very limited, and existing algorithms all suffer from various limitations. In particular, these algorithms typically rely on strong, and often infeasible assumptions such as bounded data heterogeneity or require large batch access while failing to achieve linear speedup with the number of clients. In this paper, we introduce MoTEF, a novel approach that integrates communication compression with **Mo**mentum **T**racking and **E**rror **F**eedback. MoTEF is the first algorithm to achieve an asymptotic rate matching that of distributed SGD under arbitrary data heterogeneity, hence resolving a long-standing theoretical obstacle in decentralized optimization with compressed communication. We provide numerical experiments to validate our theoretical findings and confirm the practical superiority of MoTEF.

## 1 Introduction

Decentralized machine learning approaches are increasingly popular in numerous applications such as the internet-of-things (IoT) and networked autonomous systems (Marvasti et al., 2014; Savazzi et al., 2020), primarily due to their scalability to larger datasets and systems, as well as their respect for data locality and privacy concerns. In this work, we focus on decentralized optimization techniques that operate without a central coordinator, relying solely on on-device computation and local communication with neighboring devices. This encompasses traditional scenarios like training Machine Learning models in large data centers, as well as emerging applications where computations occur directly on devices. Such a setting is preferred over centralized topology which often poses a significant bottleneck on the central node in terms of communication latency, bandwidth, and fault tolerance.

Considering the enormous size of modern Machine Learning models, classic single-node training is often impossible. Moreover, the training of large models requires a huge amount of data that does not fit the memory of a single machine. Therefore, modern training techniques heavily rely on distributed computations over a set of computation nodes/clients (Shoeybi et al., 2019; Wang et al., 2020; Ramesh et al., 2021; 2022). One of the instances of distributed training is Federated Learning (FL) (Konecnỳ et al., 2016; Kairouz et al., 2021) which has recently gathered a lot of attention. In this setting, clients, such as hospitals or owners of edge devices, collaboratively train a model on their devices while retaining their data locally.

A key issues in distributed optimization is the communication bottleneck (Seide et al., 2014; Ström, 2015) that limits the scaling properties of distributed deep learning training (Seide et al., 2014; Alistarh et al., 2017). One of the remedies to decrease communication expenses involves communication

---

*Equal contribution.

compression, where only quantized messages (with fewer bits) are exchanged between clients using compression operators. When used appropriately, contractive compressors (see Definition 1), such as Top-K, are often empirically preferable. However, the naive application of contractive compression operators might lead to divergence (Beznosikov et al., 2023). To make compression suitable for distributed training, the Error Feedback (EF) mechanism (Seide et al., 2014; Stich et al., 2018) is widely used in practice. It plays a crucial role in achieving high compression ratios.

However, most of the works analyzed EF mechanism in the centralized setting (Stich & Karimireddy, 2019; Gorbunov et al., 2020; Stich, 2020). Recent research achievements (Gao et al., 2024; Fatkhullin et al., 2024) demonstrate that in this regime properly constructed EF mechanism can handle both client drift (Mishchenko et al., 2019; Karimireddy et al., 2020b) and stochastic noise from the gradients, and can achieve near-optimal convergence rates. In the more challenging decentralized setting, a series of studies (Zhao et al., 2022; Yan et al., 2023) introduced algorithms capable of effectively managing the drift but fail to achieve a linear acceleration in parallel training, i.e. increasing the number of devices used for training does not lead to a decrease in the training time. Yau & Wai (2022) partially solved this issue under stronger assumptions achieving linear speed-up using variance reduction, but with worse dependency on the variance of the noise

Designing a method that addresses client drift while preserving linear acceleration in decentralized training has been challenging due to the complex interplay between client drift, Error Feedback mechanism and the communication topology. In our study, we introduce MoTEF, a novel method that tackles these challenges concurrently. Our primary contributions can be outlined as follows.

- We propose a novel method MoTEF that incorporates momentum tracking with compression and Error Feedback, and provably works under standard assumptions $(i)$ without imposing any data heterogeneity bounds, $(ii)$ without any impractical assumptions such as large batches, $(iii)$ with arbitrary contractive compressor, and $(iv)$ achieves linear speed-up with the number of clients $n$. We provide convergence guarantees for the general class of non-convex functions, and for the structured class of non-convex functions satisfying the Polyak-Łojasiewicz (PŁ) condition.
- We propose MoTEF-VR, a momentum-based STORM-type (Cutkosky & Orabona, 2019) variance-reduced variant of our base method that improves further the asymptotic rate of convergence.
- Finally, we provide an extensive numerical study of MoTEF demonstrating the superiority of the proposed method in practice and supporting theoretical claims.

## 1.1 RELATED WORKS

**Decentralized optimization and gradient tracking.** First works in the field studied gossip averaging procedures that are typically used to reach consensus (Kempe et al., 2003; Xiao & Boyd, 2004). Nevertheless, direct use of gossip averaging might be sub-optimal as it often results in slow convergence (Nedic & Ozdaglar, 2009). Gradient tracking (Qu & Li, 2017; Nedic et al., 2017; Koloskova et al., 2021) is one the most popular remedies to this issue. It has been widely applied to obtain faster decentralized algorithms (Sun et al., 2020; Xin et al., 2022; 2021; Li et al., 2022; Zhao et al., 2022). In this work, we follow a similar approach but perform a tracking step on momentum term instead of gradients. Takezawa et al. (2022) might be the first to analyze momentum tracking in decentralized optimization, but they do not consider communication compression.

**Momentum in distributed training.** Lately, the utilization of momentum (Polyak, 1964) has attracted attention in distributed optimization. Several works empirically showed that momentum can improve performance in distributed setting (Wang et al., 2019a; Karimireddy et al., 2020a; Das et al., 2022). Besides, it has recently been shown that the use of momentum improves convergence guarantees (Yau & Wai, 2022; Fatkhullin et al., 2024; Cheng et al., 2024; Huang et al., 2024) fully removing dependencies on data heterogeneity bounds. In this work, we follow this approach and apply the momentum technique to the more challenging decentralized setting.

**Short history of Error Feedback.** Initially, the Error Feedback mechanism was introduced as a heuristic (Seide et al., 2014) and was subsequently analyzed within a simple single-node framework (Stich et al., 2018; Karimireddy et al., 2019). The first findings in the distributed context were achieved under strong assumptions such as IID data distributions (Karimireddy et al., 2019) or bounded gradients (Cordonnier, 2018; Alistarh et al., 2018; Koloskova et al., 2019; 2020a). EF21

(Richtárik et al., 2021) stands out as the first algorithm proven to operate with any contractive compressors and under arbitrary heterogeneity, albeit failing to converge when clients are limited to using only stochastic gradients (Fatkhullin et al., 2024). Subsequently, EF21 was extended to diverse practical scenarios (Fatkhullin et al., 2021) and decentralized training (Zhao et al., 2022) improving the dependencies on some problem parameters. Recent advancements (Gao et al., 2024; Fatkhullin et al., 2024) have demonstrated that a carefully designed EF mechanism (through the control of feedback signal strength or the use of momentum) results in nearly optimal convergence guarantees in a centralized setting.

**Issues of Error Feedback in decentralized setting.** Despite having been studied in the centralized setting extensively, EF-based algorithms in the decentralized regime still fail to achieve desirable properties.

- **Strong assumptions.** Many earlier theoretical results for EF require strong assumptions, such as either the bounded gradient assumption (Koloskova et al., 2019; 2020a) or global heterogeneity bound (Lian et al., 2017; Tang et al., 2019; Lu & De Sa, 2021; Singh et al., 2021).

- **Mega batches.** Convergence of BEER algorithm(Zhao et al., 2022) requires large batches that can be costly or even infeasible in some applications. For example, in medical applications (Rieke et al., 2020) or Reinforcement Learning (Khodadadian et al., 2022; Jin et al., 2022; Mitra et al., 2023) sampling large batches is often intractable. Moreover, it has been shown that training with small batch sizes improves generalization and convergence (Wilson & Martinez, 2003; Keskar et al., 2016; Sekhari et al., 2021).

- **Suboptimal rates.** The stochastic term of several algorithms does not improve with $n$ the number of clients (Zhao et al., 2022; Yan et al., 2023), while the opposite is often desirable, and can be achieved in the centralized training setting (Fatkhullin et al., 2024; Gao et al., 2024). Other work achieves speed-up with $n$, but requires stronger smoothness assumptions and has a worse dependency on the noise variance (Yau & Wai, 2022). Moreover, (Koloskova et al., 2019; 2020a) do not achieve standard $\mathcal{O}(1/\varepsilon^2)$ convergence rate in noiseless regime.

- **Necessity of unbiased compression.** Finally, early works analyzed decentralized algorithms only for a more restricted class of *unbiased* compressors (Tang et al., 2018a; Kovalev et al., 2021). Huang & Pu (2023) modify any contractive compressor using an additional unbiased compressor following the results of (Horváth & Richtárik, 2020). This approach enables the creation of a better sequence of gradient estimators, albeit with twice the per-iteration communication cost.

In Table 1, we provide a summary of known theoretical results in decentralized training with compression that are most relevant to our work. We highlight the main issues of existing algorithms.

## 2 PROBLEM SETUP

Formally, we consider the following optimization problem

$$\min_{\mathbf{x} \in \mathbb{R}^d} \left\{ f(\mathbf{x}) := \frac{1}{n} \sum_{i=1}^n f_i(\mathbf{x}) \right\}, \tag{1}$$

where $n$ is the number of clients participating in the training, $\mathbf{x}$ are the parameters of a model, $f(\mathbf{x})$ is the global objective, and $f_i(\mathbf{x}) := \mathbb{E}_{\xi_i \sim \mathcal{D}_i}[f_i(\mathbf{x}, \xi_i)]$ is the local objective over local dataset $\mathcal{D}_i$. Throughout this work, we assume that the global function $f$ is bounded below by $f^\star > -\infty$.

In the setting of decentralized communication, the clients are restricted to communicating with their neighbors only over a certain undirected communication graph $\mathcal{G}([n], E)$. Each vertex in $[n]$ represents a client, and each edge in $E$ represents a communication link between clients. Besides, we assign a positive weight to $w_{ij}$ if there is an edge $(i, j) \in E$, and $w_{ij} = 0$ if $(i, j) \notin E$. Weights $w_{ij}$ form a mixing matrix $\mathbf{W} \in \mathbb{R}^{n \times n}$ (sometimes also called gossip or interaction matrix). The mixing matrix $\mathbf{W}$ should satisfy the following standard assumption.

**Assumption 1.** *We assume that* $\mathbf{W} \in \mathbb{R}^{n \times n}$ *is symmetric* $(\mathbf{W} = \mathbf{W}^\top)$ *and doubly stochastic* $(\mathbf{W}\mathbf{1} = \mathbf{1}, \mathbf{1}^\top \mathbf{W} = \mathbf{1}^\top)$ *matrix with eigenvalues* $1 = |\lambda_1(\mathbf{W})| > |\lambda_2(\mathbf{W})| \geq \cdots \geq |\lambda_n(\mathbf{W})|$. *We denote the spectral gap of* $\mathbf{W}$ *as*

$$\rho := 1 - |\lambda_2(\mathbf{W})| \in (0, 1]. \tag{2}$$

Table 1: Summary of convergence guarantees for decentralized methods supporting contractive compressors. **nCVX** = supports non-convex functions; **PŁ** = supports functions satisfying PŁ condition. We present the convergence in terms of $\mathbb{E}\left[\|\nabla f(\mathbf{x}_{\text{out}})\|^2\right] \leq \varepsilon^2$ and $\mathbb{E}\left[f(\mathbf{x}_{\text{out}}) - f^\star\right] \leq \varepsilon$ in PŁ regimes for specifically chosen $\mathbf{x}_{\text{out}}$. Here $F^0 := \mathbb{E}\left[f(\mathbf{x}^0) - f^*\right]$, $L$ and $\ell$ are smoothness constants, $\rho$ is a spectral gap, and $\sigma^2$ is stochastic variance bound.

| Method | Asymptotic Complexity nCVX | PŁ | Large Batches? | Extra Assumptions? |
|---|---|---|---|---|
| Choco-SGD (Koloskova et al., 2019) | $\frac{LF^0\sigma^2}{n\varepsilon^4}$ | ✗ | ✗ | Bounded Gradients $\mathbb{E}\left[\|\nabla f_i(\mathbf{x}, \xi)\|^2\right] \leq G^2$ |
| BEER (Zhao et al., 2022) | $\frac{LF^0\sigma^2}{\alpha^2\rho^3\varepsilon^4}$ | $\frac{LF^0}{\mu^2\alpha^2\rho^3\varepsilon}$ | Batch size of order $\frac{\sigma^2}{\alpha\varepsilon^2}$ | ✗ |
| CEDAS (Huang & Pu, 2023) | $\frac{LF^0\sigma^2}{n\varepsilon^4}$ | ✗ | ✗ | Additional Unbiased Compressor |
| DeepSqueeze (Tang et al., 2019) | $\frac{LF^0\sigma^2}{n\varepsilon^4}$ | ✗ | ✗ | Bounded Heterogeneity $n^{-1}\sum_i \|\nabla f_i(\mathbf{x}) - \nabla f(\mathbf{x})\|^2 \leq \zeta^2$ |
| DoCoM (Yau & Wai, 2022) | $\frac{\ell F^0\sigma^3}{n\varepsilon^3}$ | $\frac{\ell F^0\sigma^3}{\mu^2 n\varepsilon}$ | ✗ | ✗ |
| CDProxSGT (Yan et al., 2023) | $\frac{LF^0\sigma^2}{\alpha^2\rho^2\varepsilon^4}$ | ✗ | ✗ | ✗ |
| MoTEF **[This work]** | $\frac{LF^0\sigma^2}{n\varepsilon^4}$ | $\frac{LF^0\sigma^2}{\mu^2 n\varepsilon}$ | ✗ | ✗ |
| MoTEF-VR **[This work]** | $\frac{\ell F^0\sigma^2}{n\varepsilon^3}$ | ✗ | ✗ | ✗ |

The spectral gap is typically used to measure the influence of network topology in the training (Aldous & Fill, 2002; Nedić et al., 2018).

In our work, we consider algorithms combined with compressed communication. Formally, we analyze methods utilizing practically useful contractive compression operators.

**Definition 1.** *We say that a (possibly randomized) mapping $\mathcal{C}: \mathbb{R}^d \rightarrow \mathbb{R}^d$ is a contractive compression operator if for some constant $0 < \alpha \leq 1$ it holds*

$$\mathbb{E}\left[\|\mathcal{C}(\mathbf{x}) - \mathbf{x}\|^2\right] \leq (1 - \alpha)\|\mathbf{x}\|^2. \tag{3}$$

One of the classic examples of compressors satisfying (3) is Top-K (Stich et al., 2018). It acts on the input by preserving K largest by magnitude entries while zeroing the rest. The class of contractive compressors includes well-known sparsification Alistarh et al. (2018); Stich et al. (2018) and quantization (Wen et al., 2017; Bernstein et al., 2018; Horváth et al., 2022) operators. We refer to (Islamov et al., 2021; Qian et al., 2021; Beznosikov et al., 2023; Safaryan et al., 2022; Islamov et al., 2023) for more examples of contractive compressors.

In decentralized training, typically, each client receives the messages from its neighbors and transfers back to them the aggregated information. We highlight that, contrary to many prior works, our analysis supports an arbitrarily heterogeneous setting, i.e. it does not require any assumptions on the heterogeneity level, which means that local data distributions might be distant from each other. Next, we provide standard assumptions on the function class and noise model.

**Assumption 2.** *We assume that each local function $f_i$ is $L$-smooth, i.e. for all $\mathbf{x}, \mathbf{y} \in \mathbb{R}^d$, and $i \in [n]$ it holds*

$$\|\nabla f_i(\mathbf{x}) - \nabla f_i(\mathbf{y})\| \leq L\|\mathbf{x} - \mathbf{y}\|. \tag{4}$$

Next, we assume that each client has access to an unbiased gradient estimator with bounded variance.

**Assumption 3.** *We assume that we have access to a gradient oracle $\mathbf{g}^i(\mathbf{x}): \mathbb{R}^d \rightarrow \mathbb{R}^d$ for each local function $f_i$ such that for all $\mathbf{x} \in \mathbb{R}^d$ and $i \in [n]$ it holds*

$$\mathbb{E}\left[\mathbf{g}^i(\mathbf{x})\right] = \nabla f_i(\mathbf{x}), \quad \mathbb{E}\left[\|\mathbf{g}^i(\mathbf{x}) - \nabla f_i(\mathbf{x})\|^2\right] \leq \sigma^2. \tag{5}$$

It is important to mention that mini-batches are allowed as well, effectively reducing the variance by the local batch size. Nevertheless, there is no requirement for any specific (minimal) batch size, and for simplicity, we consistently assume a batch size of one.

Finally, we consider the structural class of non-convex functions satisfying Polyak-Łojasiewicz condition (Polyak, 1963). This assumption is one of the weakest conditions under which vanilla Gradient Descent converges linearly (Karimi et al., 2016).

**Assumption 4.** *We assume that the global function $f$ is $\mu$-PŁ for some $\mu > 0$, i.e. for all $\mathbf{x} \in \mathbb{R}^d$ it holds*

$$\|\nabla f(\mathbf{x})\|^2 \geq 2\mu(f(x) - f^\star). \tag{6}$$

Note that the PŁ condition is a relaxation of strong convexity, i.e. if strong convexity with parameter $\mu$ implies $\mu$-PŁ condition.

## 3 THE ALGORITHMS AND THEORETICAL ANALYSIS

In this section, we introduce our main algorithm MoTEF, summarized in Algorithm 1. MoTEF combines **Mo**mentum **T**racking with **E**rror **F**eedback to tackle the three major challenges of decentralized optimization with compression at once: client drift, stochastic noise of the gradient, and compression bias. In line 5–6 and line 9–10 we apply the EF-enhanced gossip step inspired by Zhao et al. (2022) and Koloskova et al. (2019), and in line 7 we apply the Momentum Tracking mechanism, which combines the classical Gradient Tracking method with Polyak's momentum. We will show that MoTEF is the fisrt algorithm that achieves an optimal asymptotic rate matching that of distributed SGD without any additional, and possibly impractical, assumptions.

It is well-known in the literature that the asymptotic rate of SGD cannot be improved under the standard assumptions. There is a long line of work, known as Variance Reduction, that attempts to accelerate SGD under the additional mean-squared-smoothness assumption (for which such acceleration is *crucial*) (Fang et al., 2018; Cutkosky & Orabona, 2019; Tran-Dinh et al., 2022; Wang et al., 2019b; Xu & Xu, 2022). To demonstrate the flexibility and effectiveness of our approach MoTEF, we also present a momentum-based variance-reduced variant MoTEF-VR summarized in Algorithm 2.

Next we present the theoretical analysis of MoTEF and MoTEF-VR.

### 3.1 NOTATION

Before going into details, we introduce a notation that we use throughout the paper. We stack the local parameters $\mathbf{x}_i^t$ stored at each clients into a matrix $\mathbf{X}^t := [\mathbf{x}_1^t, \ldots, \mathbf{x}_n^t] \in \mathbb{R}^{d \times n}$, and denote the average model $\bar{\mathbf{x}}^t := \frac{1}{n}\mathbf{X}^t\mathbf{1}$, where $\mathbf{1}$ is a vector of ones. Other quantities are defined similarly. To track local gradients, we define $\nabla F(\mathbf{X}^t) := [\nabla f_1(\mathbf{x}_1^t), \ldots, \nabla f_n(\mathbf{x}_n^t)] \in \mathbb{R}^{d \times n}$. Similarly we write $\widetilde{\nabla} F(\mathbf{X}^t)$ as the collection of local stochastic gradients. Finally, $\mathcal{C}_\alpha(\mathbf{X})$ denotes the contractive compression operator $\mathcal{C}_\alpha$ applied column-wise on a matrix $\mathbf{X}$, i.e. $\mathcal{C}_\alpha(\mathbf{X}) := [\mathcal{C}(\mathbf{x}_1), \ldots, \mathcal{C}(\mathbf{x}_n)] \in \mathbb{R}^{d \times n}$.

### 3.2 CONVERGENCE OF MoTEF

Now we are ready to present convergence guarantees for MoTEF. Below we summarize the convergence guarantees for Algorithm 1 in general non-convex and PŁ settings. Our analysis relies on the Lyapunov function of the form

$$\Phi^t := F^t + \frac{c_1}{n^2 L}\hat{G}^t + \frac{c_2 \tau}{nL}\widetilde{G}^t + \frac{c_3 L}{\rho^3 n \tau}\Omega_1^t + \frac{c_4 \tau}{\rho nL}\Omega_2^t + \frac{c_5 L}{\rho^3 n \tau}\Omega_3^t + \frac{c_6 \tau}{\rho nL}\Omega_4^t, \tag{7}$$

where $\{c_k\}_{k=1}^6$ are absolute constants defined in the appendix in (32)[1], $F^t := \mathbb{E}\left[f(\bar{\mathbf{x}}^t) - f^\star\right]$ represents the sub-optimality function gap, and the error terms are defined as follows

$$\hat{G}^t := \mathbb{E}\left[\|\nabla F(\mathbf{X}^t)\mathbf{1} - \mathbf{M}^t\mathbf{1}\|_{\mathrm{F}}^2\right], \quad \widetilde{G}^t := \mathbb{E}\left[\|\nabla F(\mathbf{X}^t) - \mathbf{M}^t\|_{\mathrm{F}}^2\right], \quad \Omega_1^t := \mathbb{E}\left[\|\mathbf{H}^t - \mathbf{X}^t\|_{\mathrm{F}}^2\right]$$

$$\Omega_2^t := \mathbb{E}\left[\|\mathbf{G}^t - \mathbf{V}^t\|_{\mathrm{F}}^2\right], \quad \Omega_3^t := \mathbb{E}\left[\|\mathbf{X}^t - \bar{\mathbf{x}}^t\mathbf{1}^T\|_{\mathrm{F}}^2\right] \tag{8}$$

$$\Omega_4^t := \mathbb{E}\left[\|\mathbf{V}^t - \bar{\mathbf{v}}^t\mathbf{1}^T\|_{\mathrm{F}}^2\right], \quad \Omega_5^t := \mathbb{E}\left[\|\bar{\mathbf{v}}^t\|^2\right].$$

---

[1]To find a suitable choice of constants we use Symbolic Math Toolbox in MATLAB (Inc., 2023). Our code can be found at https://anonymous.4open.science/r/dec-symb-verification.

| **Algorithm 1** MoTEF | **Algorithm 2** MoTEF-VR |
|---|---|
| 1: **Input:** $\mathbf{X}^0 = \mathbf{x}^0 \mathbf{1}^\top, \mathbf{G}^0, \mathbf{H}^0, \mathbf{V}^0, \gamma, \eta, \lambda,$ | 1: **Input:** $\mathbf{X}^0 = \mathbf{x}^0 \mathbf{1}^\top, \mathbf{G}^0, \mathbf{H}^0, \mathbf{V}^0, \gamma, \eta, \lambda,$ |
| 2: $\mathcal{C}_\alpha$ | 2: $\mathcal{C}_\alpha$ |
| 3: **for** $t = 0, 1, 2, \ldots$ **do** | 3: **for** $t = 0, 1, 2, \ldots$ **do** |
| 4: $\quad \mathbf{X}^{t+1} = \mathbf{X}^t + \gamma \mathbf{H}^t(\mathbf{W} - \mathbf{I}) - \eta \mathbf{V}^t$ | 4: $\quad \mathbf{X}^{t+1} = \mathbf{X}^t + \gamma \mathbf{H}^t(\mathbf{W} - \mathbf{I}) - \eta \mathbf{V}^t$ |
| 5: $\quad \mathbf{Q}_h^{t+1} = \mathcal{C}_\alpha(\mathbf{X}^{t+1} - \mathbf{H}^t)$ | 5: $\quad \mathbf{Q}_h^{t+1} = \mathcal{C}_\alpha(\mathbf{X}^{t+1} - \mathbf{H}^t)$ |
| 6: $\quad \mathbf{H}^{t+1} = \mathbf{H}^t + \mathbf{Q}_h^{t+1}$ | 6: $\quad \mathbf{H}^{t+1} = \mathbf{H}^t + \mathbf{Q}_h^{t+1}$ |
| 7: $\quad \mathbf{M}^{t+1} = (1 - \lambda)\mathbf{M}^t + \lambda \widetilde{\nabla} F(\mathbf{X}^{t+1})$ | 7: $\quad \mathbf{M}^{t+1} = \widetilde{\nabla} F(\mathbf{X}^{t+1}, \Xi^{t+1})$ |
| 8: $\quad \mathbf{V}^{t+1} = \mathbf{V}^t + \gamma \mathbf{G}^t(\mathbf{W} - \mathbf{I}) + \mathbf{M}^{t+1} - \mathbf{M}^t$ | 8: $\quad + (1 - \lambda)(\mathbf{M}^t - \widetilde{\nabla} F(\mathbf{X}^t, \Xi^{t+1}))$ |
| 9: $\quad \mathbf{Q}_g^{t+1} = \mathcal{C}_\alpha(\mathbf{V}^{t+1} - \mathbf{G}^t)$ | 9: $\quad \mathbf{V}^{t+1} = \mathbf{V}^t + \gamma \mathbf{G}^t(\mathbf{W} - \mathbf{I}) + \mathbf{M}^{t+1} - \mathbf{M}^t$ |
| 10: $\quad \mathbf{G}^{t+1} = \mathbf{G}^t + \mathbf{Q}_g^{t+1}$ | 10: $\quad \mathbf{Q}_g^{t+1} = \mathcal{C}_\alpha(\mathbf{V}^{t+1} - \mathbf{G}^t)$ |
| | 11: $\quad \mathbf{G}^{t+1} = \mathbf{G}^t + \mathbf{Q}_g^{t+1}$ |

Our theory relies on the descent of the Lyapunov function $\Phi^t$ introduced above.

**Lemma 1** (Descent of the Lyapunov function). *Let Assumptions 2 and 3 hold. Then there exist absolute constants $c_\gamma, c_\lambda, c_\eta$, and $\tau \leq 1$ such that if we set stepsizes $\gamma = c_\gamma \alpha \rho, \lambda = c_\lambda \alpha \rho^3 \tau, \eta = c_\eta L^{-1} \alpha \rho^3 \tau$ such that the Lyapunov function $\Phi^t$ decreases as*

$$\Phi^{t+1} \leq \Phi^t - \frac{c_\eta \alpha \rho^3 \tau}{2L} \mathbb{E}\left[\|\nabla f(\bar{\mathbf{x}}^t)\|^2\right] + \frac{c_\lambda^2 c_1 \alpha^2 \rho^6}{nL} \tau^2 \sigma^2 + \tau^3 \sigma^2 \left(3c_4 \rho + c_2 \alpha \rho^2 + \frac{3c_6}{c_\gamma}\right) \frac{2c_\lambda^2 \alpha \rho^4}{L}. \quad (9)$$

Using the above descent of the Lyapunov function, we demonstrate the convergence guarantees for MoTEF.

**Theorem 1** (Convergence of MoTEF). *Let Assumptions 2 and 3 hold. Then there exist absolute constants $c_\gamma, c_\lambda, c_\eta$, and some $\tau \leq 1$ such that if we set stepsizes $\gamma = c_\gamma \alpha \rho, \lambda = c_\lambda \alpha \rho^3 \tau, \eta = c_\eta L^{-1} \alpha \rho^3 \tau$, and choosing the initial batch size $B_{\mathrm{init}} \geq \lceil \frac{LF^0}{\sigma^2} \rceil$, then after at most*

$$T = \mathcal{O}\left(\frac{\sigma^2}{n\varepsilon^4} + \frac{\sigma}{\alpha \rho^{5/2} \varepsilon^3} + \frac{1}{\alpha \rho^3 \varepsilon^2}\right) LF^0 \quad (10)$$

*iterations of Algorithm 1 it holds $\mathbb{E}\left[\|\nabla f(\mathbf{x}_{\mathrm{out}})\|^2\right] \leq \varepsilon^2$, where $\mathbf{x}_{\mathrm{out}}$ is chosen uniformly at random from $\{\bar{\mathbf{x}}_0, \ldots, \bar{\mathbf{x}}_{T-1}\}$, and $\mathcal{O}$ suppresses absolute constants.*

**Remark 2.** *Note that using a large initial batch size $B_{\mathrm{int}}$ is not required for convergence of MoTEF. If we set $B_{\mathrm{init}} = 1$, the above theorem still holds by replacing $F^0$ by $\Phi^0$.*

We observe that the use of momentum in MoTEF allows us to improve convergence guarantees over BEER. Indeed, Algorithm 1 achieves optimal asymptotic complexity[2] with a desirable linear speed-up with the number of clients $n$. Moreover, MoTEF provably converges for any batch size in contrast to BEER.

**Discussion of the convergence rate.** In the stochastic regime ($\sigma^2 > 0$), we note that the asymptotically dominating term is $\mathcal{O}(\sigma^2/n\varepsilon^4)$, which is independent of the spectral gap and compression rate and is optimal in all problem parameters (Arjevani et al., 2023). To the best of our knowledge, MoTEF is the first decentralized algorithm incorporating contractive compressors that achieves it under Assumptions 2 and 3 without data heterogeneity restrictions. In the deterministic regime ($\sigma^2 = 0$), the convergence rate becomes $\mathcal{O}(1/\alpha\rho^3\varepsilon^2)$. This is optimal in $\alpha$ and $\varepsilon$ (Huang et al., 2022) and sub-optimal in $\rho$. We would like to highlight that having a sub-optimal dependency on the spectral gap $\rho$ is a well-known challenge in the theoretical analysis of the gradient tracking mechanism (Koloskova et al., 2021). Besides, the same sub-optimal $1/\rho^3$ dependency is also observed in the analysis of BEER algorithm. It is an active research direction to either improve the convergence analysis of gradient tracking to obtain better $\rho$ dependence (Koloskova et al., 2021) or to design more sophisticated tracking mechanisms that might achieve better rates (Di et al., 2022). Either way, it involves significantly more complicated analyses and we defer these to future work. We also point out that it is unclear whether the $1/\rho^3$ dependence is inherent to the tracking mechanism or is an artifact

---

[2]This means the regime when $\varepsilon \to 0$.

of the analysis. In Section 4 we provide numerical evidence showing that MoTEF might be much less sensitive to $\rho$ than the theoretical analysis suggests. A more detailed discussion of the results of previous works is deferred to Appendix A.

Now we derive convergence guarantees of MoTEF for the class of functions satisfying Assumption 4.

**Theorem 2** (Convergence of MoTEF). *Let Assumptions 2 to 4 hold. Then there exist absolute constants $c_\gamma, c_\lambda, c_\eta$, and some $\tau \leq 1$ such that if we set stepsizes $\gamma = c_\gamma \alpha \rho, \lambda = c_\lambda \alpha \rho^3 \tau, \eta = c_\eta L^{-1} \alpha \rho^3 \tau$, and choosing the initial batch size $B_{\text{init}} \geq \lceil \frac{LF^0}{\sigma^2} \rceil$, then after at most*

$$T = \widetilde{\mathcal{O}} \left( \frac{L\sigma^2}{\mu^2 n \varepsilon} + \frac{L\sigma}{\alpha \rho^{5/2} \mu^{3/2} \varepsilon^{1/2}} + \frac{L}{\mu \alpha \rho^3} \right) \tag{11}$$

*iterations of Algorithm 1 it holds $\mathbb{E}\left[ f(\mathbf{x}^T) - f^* \right] \leq \varepsilon$, and $\widetilde{\mathcal{O}}$ suppresses absolute constants and poly-logarithmic factors.*

**Remark 3.** *Note that using a large initial batch size $B_{\text{int}}$ is not required for convergence of MoTEF. If we set $B_{\text{init}} = 1$, the above theorem still holds by replacing $F^0$ by $\Phi^0$, which is hidden in the logarithmic terms.*

Contrary to BEER, we demonstrate that the asymptotic rate of MoTEF in the PŁ setting improves with $n$ and does not require large batches. To the best of our knowledge, MoTEF is the first decentralized algorithm that supports contractive compressors and achieves linear speed-up with $n$ under Assumptions 2 to 4. Moreover, we highlight that in the noiseless regime MoTEF converges linearly as expected. Another momentum-based algorithm DoCom was analyzed under more restricted Assumption 5 only. Therefore, its applicability in this setting is not known. Besides, DoCoM achieves linear speed-up with $n$, but with sub-optimal dependency on the noise variance $\sigma^2$.

## 3.3 CONVERGENCE OF MoTEF-VR

We demonstrated that MoTEF achieves an asymptotic complexity of distributed SGD under Assumptions 2 and 3, and this result cannot be improved. However, if we consider strengthening of Assumption 2, the mean-squared-smoothness Assumption 5, then further acceleration on the stochastic term might be achieved via variance reduction. We emphasize that Assumption 5 is the standard assumption made for variance reduction, and is the key one for circumventing existing lower bounds on stochastic methods (Fang et al., 2018; Cutkosky & Orabona, 2019; Tran-Dinh et al., 2022; Wang et al., 2019b; Xu & Xu, 2022).

**Assumption 5.** *We assume that each local function $f_i$ is $\ell$-mean-squared-smooth, i.e. for all $\mathbf{x}, \mathbf{y} \in \mathbb{R}^d, i \in [n]$, it holds*

$$\mathbb{E}_\xi \left[ \|\nabla f_i(\mathbf{x}, \xi) - \nabla f_i(\mathbf{y}, \xi)\|^2 \right] \leq \ell^2 \|\mathbf{x} - \mathbf{y}\|. \tag{12}$$

In MoTEF-VR, instead of a simple momentum term, each client now maintains a momentum-based variance reduction term, similar to the STORM estimator (Cutkosky & Orabona, 2019). The algorithm also maintains a momentum parameter $\lambda$, and it turns out that the additional variance reduction terms and Assumption 5 allow us to set the momentum parameter more aggressively, leading to an improved convergence rate.

**Theorem 3** (Convergence of MoTEF-VR). *Let Assumptions 3 and 5 hold. Then there exists absolute constants $c_\gamma, c_\lambda, c_\eta$ and some $\tau < 1$ such that if we stepsizes $\gamma = c_\gamma \alpha \rho, \lambda = c_\lambda n^{-1} \alpha^2 \rho^6 \tau^2, \eta = c_\eta \ell^{-1} \alpha \rho^3 \tau$, and initial batch size $B_{\text{init}} \geq \lceil \frac{\sigma^2}{LF^0 \alpha \rho^3} \rceil$, then after at most*

$$T = \mathcal{O} \left( \frac{\sigma}{n \varepsilon^3} + \frac{\sigma^{2/3}}{n^{2/3} \alpha^{1/3} \rho^{2/3} \varepsilon^{8/3}} + \frac{1}{\alpha \rho^3 \varepsilon^2} \right) \ell F^0 \tag{13}$$

*iterations of Algorithm 2 it holds $\mathbb{E}\left[ \|\nabla f(\mathbf{x}_{\text{out}})\|^2 \right] \leq \varepsilon^2$, where $\mathbf{x}_{\text{out}}$ is chosen uniformly at random from $\{\bar{\mathbf{x}}_0, \cdots, \bar{\mathbf{x}}_{T-1}\}$, and $\mathcal{O}$ suppresses absolute constants and poly-logarithmic factors.*

**Remark 4.** *Note that using a large initial batch size $B_{\text{init}}$ is not required for convergence of MoTEF-VR. If we set $B_{\text{init}} = 1$, the above theorem still holds replacing $F^0$ by $\Psi^0$.*

Compared to MoTEF, MoTEF-VR achieves an improved asymptotic rate. Moreover, all stochastic terms (the ones with $\sigma$) have a speed-up with $n$ in contrast to the convergence of DoCoM, where only asymptotic term improves with $n$.

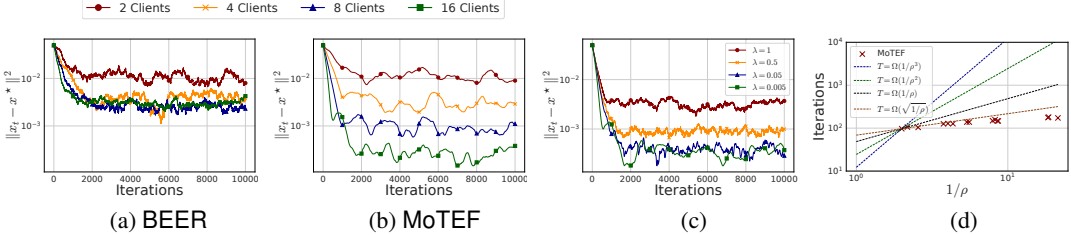

Figure 1: (a) BEER with different number of clients $n$; (b) MoTEF with different number of clients $n$; (c) MoTEF with different momentum parameter $\lambda$. MoTEF's error decreases as the number of clients increases, while the error of BEER does not. The error of MoTEF increases as the momentum parameter increases. In all cases, we set $d = 20, \zeta = 10, \sigma = 10$, and apply Top-K compressor with $\alpha = {}^{K}/{}_{d} = 0.1$. We fix the parameters $\gamma = 0.1, \eta = 0.0005, \lambda = 0.005$, and $n = 16$, if the opposite is not stated. (d) The number of iterations for MoTEF to reach an error of $10^{-3}$, as compared to the theoretical prediction $\mathcal{O}({}^{1}/{}_{\rho^{3}})$. We see that the convergence of MoTEF is much less sensitive to $\rho$ than the theoretical prediction.

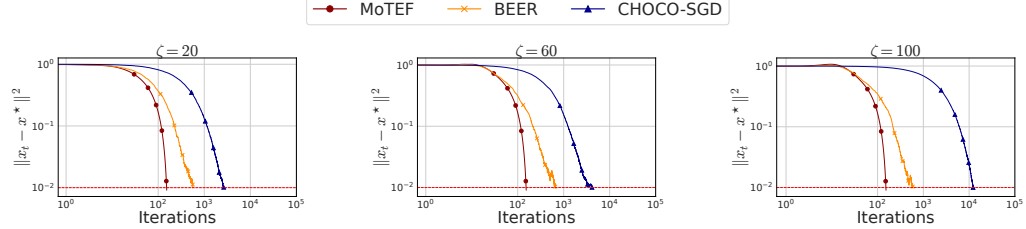

Figure 2: Performance of MoTEF, BEER and CHOCO-SGD with varying data heterogeneity $\zeta$ and fixed noise level $\sigma = 5$. We see that MoTEF outperforms BEER and CHOCO-SGD in all cases, and is not affected by the data heterogeneity, while CHOCO-SGD's performance degrades as $\zeta$ increases. We set $d = 20, n = 4$ and apply Top-K compressor with $\alpha = {}^{K}/{}_{d} = 0.1$. We set the target error to be 0.01.

We point out that MoTEF-VR applies the STORM mechanism locally to achieve the variance reduction effect. STORM is specifically designed for non-convex optimization problems, and its convergence rate in the more structured class of functions satisfying Assumption 4 is still unclear in the literature (Cutkosky & Orabona, 2019; Xu & Xu, 2022) even for the simplest centralized SGD setting. In this work, we also do not consider the rate of MoTEF-VR under the PŁ condition.

# 4 NUMERICAL EXPERIMENTS

In this section, we complement the theoretical results on the convergence of Algorithm 1 with numerical evaluations.

## 4.1 SYNTHETIC LEAST SQUARES PROBLEM

We first consider a simple synthetic least squares problem to demonstrate some of the important theoretical properties of Algorithm 1. This problem is designed by Koloskova et al. (2020b) and studied in (Gao et al., 2024). For each client $i$, $f_i(\mathbf{x}) \coloneqq \frac{1}{2} \|\mathbf{A}_i \mathbf{x} - \mathbf{b}_i\|^2$, where $\mathbf{A}_i^2 \coloneqq i^2/n \cdot \mathbf{I}_d$ and each $\mathbf{b}_i$ is sampled from $\mathcal{N}(0, \zeta^2/i^2 \mathbf{I}_d)$ for some parameter $\zeta$ which controls the gradient dissimilarity of the problem (Koloskova et al., 2020b). It's easy to see that when $\zeta = 0, \nabla f_i(\mathbf{x}^\star) = 0, \forall i$. We add Gaussian noise to the gradients to control the stochastic level $\sigma^2$ of the gradient. We use the ring network topology for the synthetic experiment unless stated otherwise.[3] We run our experiments on AMD EPYC 9554 64-Core Processor.

**Increasing the number of nodes.** In Figure 1-(a-b) we study the effect of increasing the number of nodes on the convergence of Algorithm 1. A crucial property of Algorithm 1 is that its convergence rate provably improves linearly with the number of nodes, which BEER does not possess. Here we fix a small stepsize and investigate the error that Algorithm 1 achieves with an increasing number of nodes. We observe that the error decreases linearly with the number of nodes, which is consistent with the theoretical results, while for the error of BEER it is not the case.

---

[3]The code to reproduce our synthetic experiment is available at https://anonymous.4open.science/r/decentralized-exp-A3C6

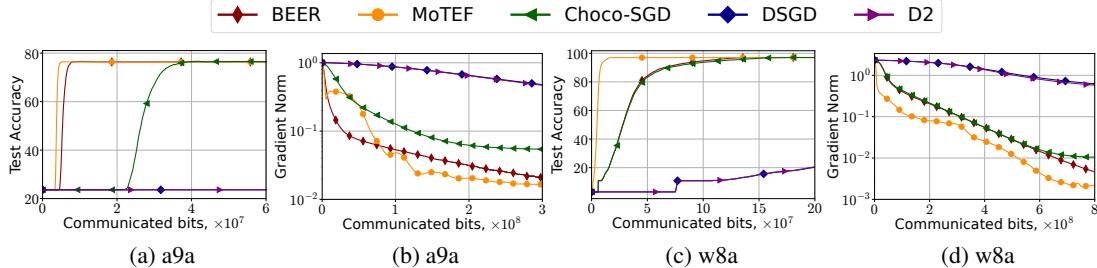

Figure 3: Comparison of MoTEF, BEER, Choco-SGD, DSGD, D2 in terms of communication complexity on logistic regression with non-convex regularization on ring topology with batch size 5 and gsgd$_b$ compressor. We observe that MoTEF outperforms other algorithms in terms of both test accuracy and gradient norm.

**Effect of the momentum parameter.** In Figure 1-(c) we investigate the effect of the momentum parameter $\lambda$. In particular, how it affects the convergence in the noisy regime. Our theoretical analysis suggests that the momentum parameter $\lambda \propto \eta$ is crucial for the convergence of MoTEF. We observe that the error increases as the momentum parameter increases. Note that when $\lambda = 1$, we recover BEER which is known to not converge with the presence of noise in the local gradients, which our experiment confirms.

**Effect of changing heterogeneity.** In Figure 2 we investigate the effect of changing data heterogeneity $\zeta$ on the performance of MoTEF, BEER, and Choco-SGD. The hyperparameters were tuned; the detailed description is given in Appendix D.1. We observe that MoTEF outperforms other algorithms and is not affected by the changing $\zeta$. BEER is also not affected by the changing $\zeta$, while CHOCO-SGD's performance degrades as $\zeta$ increases. This is consistent with the theoretical results.

**Effect of communication topologies** In Figure 1-(d) we investigate the effect of the spectral gap $\rho$ on the convergence of MoTEF. We set $\sigma^2 = 0$ since the optimization term is most affected by $\rho$ in the analysis. We detail the setup of the communication network in Appendix D.2. While the theory suggests that there might be a $1/\rho^3$ dependence, our experiment shows that the convergence of MoTEF is much less sensitive to $\rho$. Future research is needed to understand the discrepancy between the theory and the practice of the tracking mechanisms.

## 4.2 NON-CONVEX LOGISTIC REGRESSION

Following (Khirirat et al., 2023; Makarenko et al., 2023; Islamov et al., 2024) we compare algorithms on logistic regression problem with non-convex regularization[4]

$$\min_{\mathbf{x} \in \mathbb{R}^d} \frac{1}{n} \sum_{i=1}^n f_i(\mathbf{x}) + \lambda \sum_{j=1}^d \frac{x_j^2}{1+x_j^2}, \quad f_i(\mathbf{x}) \coloneqq \frac{1}{m} \sum_{j=1}^m \log(1 + \exp(-b_{ij}\mathbf{a}_{ij}^\top \mathbf{x})), \tag{14}$$

where $\{b_{ij}, \mathbf{a}_{ij}\}_{j=1}^m$ is a local dataset. We set $\lambda = 0.05, n = 100$ and use LibSVM datasets (Chang & Lin, 2011). We do not shuffle datasets to have a more heterogeneous setting. Besides, each dataset is equally distributed among all clients. In all experiments on logistic regression, we use gsgd$_b$ compressor (Alistarh et al., 2017) with $b = 5$. More details of this experiments are given in Appendix D.

**Comparison against other methods.** We compare BEER (Zhao et al., 2022), Choco-SGD (Koloskova et al., 2019), DSGD (Alistarh et al., 2017), and D2 (Tang et al., 2018b) algorithms with MoTEF on ring topology. Detailed description is given in Appendix D.4. For each algorithm, we fine-tune all stepsizes to achieve better convergence. According to the results in Figure 3, we observe that MoTEF outperforms other algorithms in terms of communication complexity in both cases, when the convergence is measured by training gradient norm and test accuracy. In Figure 7, we additionally compare MoTEF against CEDAS (Huang & Pu, 2023).

**Robustness to communication topology.** Next, we study the effect of the network topology on the convergence of MoTEF. We run experiments for ring, star, grid, Erdös-Rènyi ($p = 0.2$ and $p = 0.5$) topologies. Note the spectral gaps of these networks $0.012, 0.049, 0.063, 0.467, 0.755$ correspondingly. The hyperparameters of algorithms are given in Appendix D.3. Despite the

---

[4]Our implementation is based on open-source code from (Zhao et al., 2022) `https://github.com/liboyue/beer` and is available at `https://anonymous.4open.science/r/MoTEF-0DCF`.

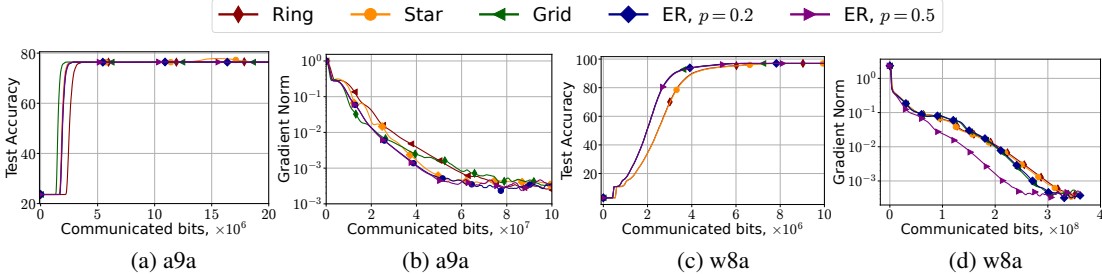

Figure 4: Performance of MoTEF changing of network topology tested on logistic regression with non-convex regularization. We set $n = 40, \lambda = 0.05$, and batch size 100. We observe that MoTEF is very robust against changing network topologies for practical problems.

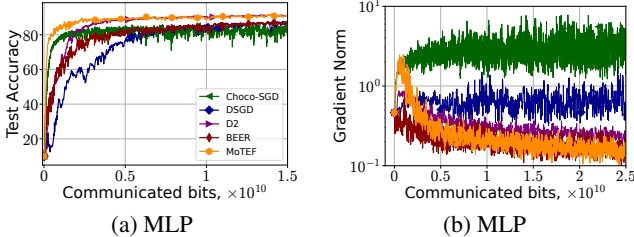

Figure 5: Comparison of MoTEF, BEER, Choco-SGD, DSGD, D2 in terms of communication complexity on training MLP with 1 hidden layer. We observe that MoTEF outperforms the other methods.

theoretical analysis showing a strong dependence on $\rho$, in Figure 4 we demonstrate that convergence MoTEF is not affected much by the change in spectral gap. These results demonstrate the robustness of MoTEF to the change of network topology in practice.

**Training of MLP.** Finally, we consider training MLP on MNIST dataset (Deng, 2012) with 1 hidden layer of size 32. We present the results in Figure 5. We observe that MLP trained with MoTEF and BEER achieve similar gradient norm, but MoTEF is much faster in accuracy metric showing the advantage from using momentum tracking. In addition, we provide the results of training a CNN model in Appendix D.5.

## 5 CONCLUSION AND OUTLOOK

In this work, we address a critical challenge in decentralized stochastic non-convex optimization with communication compression, that is, achieving the optimal asymptotic rate of $\mathcal{O}(\sigma^2/n\varepsilon^4)$ matching that of the distributed SGD under the standard assumptions and without any impractical assumptions, such as bounded data heterogeneity or access to large batches. We propose a new algorithm, MoTEF, incorporating momentum tracking and Error Feedback, and prove that it achieves this goal. We also extend the framework to MoTEF-VR and show that it achieves the variance-reduced rates under standard variance reduction assumption. We support our theoretical findings with an extensive experimental study.

The tracking mechanism plays a critical role in our algorithmic design, and a well-known challenge in these tracking mechanism is that it induces worse dependence on the spectral gap of the network. However, our preliminary numerical experiment shows that MoTEF might be much less sensitive to the spectral gap than what the theory predicts. We believe that future work can look into this aspect and either improve our analysis or design even better tracking mechanisms. In our study, we focus only on compressed communication while there are many approaches such as performing several local steps (Mishchenko et al., 2022b; Gorbunov et al., 2021) or asynchronous communication (Islamov et al., 2024; Mishchenko et al., 2022a) that might be useful. We also note that some recent works attempt to improve the dependencies on the smoothness parameters for variants of Error Feedback algorithms (Richtárik et al., 2024), where each local objective is assumed to be $L_i$-smooth, and a more careful analysis of the method gives a dependency on the average-smoothness $\bar{L} = n^{-1} \sum_{i=1}^{n} L_i$ instead of the maximum smoothness $L = \max_{i \in [n]} L_i$. Therefore, combining the aforementioned research directions with our proof techniques might lead to more improved results. We defer the exploration of these possible extensions to future research endeavors.

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

## A  EXTENDED RELATED WORK

In this section, we provide an additional discussion on the related works on decentralized optimization specifically focusing on the dependency of the deterministic optimization term on the spectral gap $\rho$.

In the non-compressed, strongly convex, and deterministic regime, Kovalev et al. (2020) and Mishchenko et al. (2022b) achieve optimal $\widetilde{\mathcal{O}}(\sqrt{L/\mu\rho})$, i.e. the dependency on the spectral $\mathcal{O}(1/\sqrt{\rho})$. Kovalev et al. (2020) proposed an algorithm APAPC based on the Chebyshev acceleration while Mishchenko et al. (2022b) boosts the convergence through incorporating multiple local steps. We highlight that both algorithms do not impose any bounds on the data heterogeneity. Later, the results of Mishchenko et al. (2022b) were extended to the stochastic regime (Guo et al., 2023) beyond strongly convex functions with a linear speed-up in $n$. However, the dependency on $\rho$ in the stochastic regime worsened to $\mathcal{O}(1/\rho^2)$. Liu et al. (2021); Li et al. (2021) achieved $\widetilde{\mathcal{O}}(L/\mu\rho)$ convergence rate but with the use of a stricter class of unbiased compression operators and in the full-batch regime. Zhang et al. (2023) proposed an algorithm called COLD that attains $\widetilde{\mathcal{O}}(L/\mu + \frac{1}{\alpha^2\rho})$ convergence rate in the deterministic strongly convex regime with contractive compressors. BEER algorithm (Zhao et al., 2022) achieves the rate that depends on $\frac{1}{\rho^3}$ similarly to the rate of MoTEF but under unrealistic large batch requirement. DoCoM algorithm attains the linear speed-up with $n$ in the stochastic regime but has worse $\frac{1}{\rho^4}$ dependency on the spectral gap. Both DeepSqueeze (Tang et al., 2018a) and Choco-SGD (Koloskova et al., 2019) achieve $\frac{1}{\rho^2}$ but under bounded data heterogeneity assumptions.

To summarize, to the best of our knowledge, there is no work in the most general setting considered in this work, namely, stochastic non-convex decentralized optimization with contractive compression under arbitrary data heterogeneity, that achieves better dependency on the spectral gap $\rho$. Most of the works make additional assumptions with a possible improvement of the rate w.r.t. $\rho$, however, the question remains if the results are transferable to the considered setting. Therefore, additional effort is needed to either improve the convergence guarantees of MoTEF with a more involved analysis using an enhanced tracking mechanism or show the lower bound that the optimal $\frac{1}{\sqrt{\rho}}$ dependency is not achievable in the worst case.

### A.1  INTUITION BEHIND MoTEF ALGORITHM DESIGN

Designing an algorithm with strong convergence guarantees without imposing assumptions on the problem or data is complicated. In MoTEF we incorporate three main ingredients to make it converge faster under arbitrary data heterogeneity. In particular, the combination of EF21-type Error Feedback (Richtárik et al., 2021) and Gradient Tracking mechanisms is the key factor in getting rid of the influence of data heterogeneity. We emphasize that not using one of them would lead to restrictions on the data heterogeneity. Indeed, EF21 is known to remove such dependencies in centralized training while the GT mechanism is essential in decentralized learning (Koloskova et al., 2021). Nonetheless, EF21 does not handle the error coming from stochastic gradients and momentum is known to be one of the remedies to it (Fatkhullin et al., 2024).

## B  MISSING PROOF FOR MoTEF

We recall the notation we use to prove convergence of MoTEF:

$$\hat{G}^t := \mathbb{E}\left[\left\|\nabla F(\mathbf{X}^t)\mathbf{1} - \mathbf{M}^t\mathbf{1}\right\|^2\right]$$

$$\widetilde{G}^t := \sum_{i=1}^{n} \mathbb{E}\left[\left\|\nabla f_i(\mathbf{x}_i^t) - \mathbf{m}_i^t\right\|^2\right] = \mathbb{E}\left[\left\|\nabla F(\mathbf{X}^t) - \mathbf{M}^t\right\|_F^2\right]$$

$$\Omega_1^t := \mathbb{E}\left[\left\|\mathbf{H}^t - \mathbf{X}^t\right\|_F^2\right]$$

$$\Omega_2^t := \mathbb{E}\left[\left\|\mathbf{G}^t - \mathbf{V}^t\right\|_F^2\right]$$

$$\Omega_3^t := \mathbb{E}\left[\left\|\mathbf{X}^t - \bar{\mathbf{x}}^t\mathbf{1}^\top\right\|_F^2\right]$$

$$\Omega_4^t := \mathbb{E}\left[\left\|\mathbf{V}^t - \bar{\mathbf{v}}^t\mathbf{1}^\top\right\|_F^2\right]$$

$$\Omega_5^t := \mathbb{E}\left[\left\|\bar{\mathbf{v}}^t\right\|^2\right].$$

Moreover, $F^t := \mathbb{E}\left[f(\bar{\mathbf{x}}^t)\right] - f^\star$. Let us define $\mathbf{\Omega}^t := [\hat{G}^t, \widetilde{G}^t, \Omega_1^t, \Omega_2^t, \Omega_3^t, \Omega_4^t]^\top$. In addition, we denote $\widetilde{\nabla}F(\mathbf{X}^t) := [\mathbf{g}^1(\mathbf{x}_i^t), \ldots, \mathbf{g}^n(\mathbf{x}_n^t)] \in \mathbb{R}^{d \times n}$ a matrix that contains local stochastic gradients. We denote $C := \sigma_{\max}^2(\mathbf{W} - \mathbf{I}) \leq 4$.

**Lemma 5** (Lemma B.2 from (Zhao et al., 2022)). *Let $\mathbf{W}$ be a mixing matrix with a spectral gap $\rho$. Then for any matrix $\mathbf{X} \in \mathbb{R}^{d \times n}$ and $\bar{\mathbf{x}} = \frac{1}{n}\mathbf{X}\mathbf{1}$ we have*

$$\|\mathbf{X}\mathbf{W} - \bar{\mathbf{x}}\mathbf{1}^\top\|_F^2 \leq (1-\rho)\|\mathbf{X} - \bar{\mathbf{x}}\mathbf{1}^\top\|_F^2. \tag{15}$$

*Moreover, for any $\gamma \in (0,1]$ the matrix $\widetilde{\mathbf{W}} = \mathbf{I} + \gamma(\mathbf{W} - \mathbf{I})$ has a spectral gap at least $\gamma\rho$.*

**Lemma 6.** *The iterates of Algorithm 1 satisfy*

$$\bar{\mathbf{v}}^{t+1} = \frac{1}{n}\mathbf{M}^{t+1}\mathbf{1}, \tag{16}$$

*and*

$$\bar{\mathbf{x}}^{t+1} = \bar{\mathbf{x}}^t - \frac{\eta}{n}\mathbf{M}^t\mathbf{1}. \tag{17}$$

*Proof.* By induction, we can show that $\bar{\mathbf{v}}^t = \frac{1}{n}\mathbf{M}^t\mathbf{1}$, if we initialize $\mathbf{V}^0 = \mathbf{M}^0$. Indeed, we have

$$\begin{aligned}
\bar{\mathbf{v}}^{t+1} &= \frac{1}{n}\mathbf{V}^{t+1}\mathbf{1} \\
&= \frac{1}{n}\mathbf{V}^t\mathbf{1} + \frac{1}{n}\gamma\mathbf{G}^t(\mathbf{W} - \mathbf{I})\mathbf{1} + \frac{1}{n}(\mathbf{M}^{t+1} - \mathbf{M}^t)\mathbf{1} \\
&= \frac{1}{n}\mathbf{V}^t\mathbf{1} + \frac{1}{n}(\mathbf{M}^{t+1} - \mathbf{V}^t)\mathbf{1} \\
&= \frac{1}{n}\mathbf{M}^{t+1}\mathbf{1}.
\end{aligned}$$

Therefore, we have

$$\begin{aligned}
\bar{\mathbf{x}}^{t+1} &= \bar{\mathbf{x}}^t + \frac{\gamma}{n}\mathbf{H}^t(\mathbf{W} - \mathbf{I})\mathbf{1} - \frac{\eta}{n}\mathbf{V}^t\mathbf{1} \\
&= \bar{\mathbf{x}}^t - \eta\bar{\mathbf{v}}^t = \bar{\mathbf{x}}^t - \frac{\eta}{n}\mathbf{M}^t\mathbf{1}.
\end{aligned}$$

$\square$

## B.1 GENERAL NON-CONVEX SETTING.

**Lemma 7.** *Assume that Assumption 2 holds. Then we have the following descent on $F^t$*

$$F_{t+1} \leq F_t - \frac{\eta}{2}\mathbb{E}\left[\|\nabla f(\bar{\mathbf{x}}^t)\|^2\right] + \frac{\eta}{n^2}\hat{G}^t + \frac{\eta L^2}{n}\Omega_3^t - (-\eta/2 - \eta^2 L/2)\Omega_5^t. \tag{18}$$

*Proof.* Using smoothness we get

$$
\begin{aligned}
F_{t+1} &\leq F_t - \eta \mathbb{E}\left[\langle \nabla f(\bar{\mathbf{x}}^t), \bar{\mathbf{v}}^t \rangle\right] + \frac{\eta^2 L}{2} \mathbb{E}\left[\|\mathbf{v}^t\|^2\right] \\
&= F_t - \frac{\eta}{2} \mathbb{E}\left[\|\nabla f(\bar{\mathbf{x}}^t)\|^2\right] + \frac{\eta}{2} \mathbb{E}\left[\|\nabla f(\bar{\mathbf{x}}^t) - \bar{\mathbf{v}}^t\|^2\right] - (-\eta/2 - \eta^2 L/2)\mathbb{E}\left[\|\bar{\mathbf{v}}^t\|^2\right] \\
&= F_t - \frac{\eta}{2} \mathbb{E}\left[\|\nabla f(\bar{\mathbf{x}}^t)\|^2\right] + \frac{\eta}{2} \mathbb{E}\left[\left\|\frac{1}{n}\nabla F(\bar{\mathbf{x}}^t)\mathbf{1} - \frac{1}{n}\mathbf{M}^t\mathbf{1}\right\|^2\right] - (-\eta/2 - \eta^2 L/2)\mathbb{E}\left[\|\bar{\mathbf{v}}^t\|^2\right] \\
&\leq F_t - \frac{\eta}{2} \mathbb{E}\left[\|\nabla f(\bar{\mathbf{x}}^t)\|^2\right] + \eta \mathbb{E}\left[\left\|\frac{1}{n}\nabla F(\mathbf{X}^t)\mathbf{1} - \frac{1}{n}\mathbf{M}^t\mathbf{1}\right\|^2\right] \\
&\quad + \eta \mathbb{E}\left[\left\|\frac{1}{n}\nabla F(\bar{\mathbf{x}}^t)\mathbf{1} - \frac{1}{n}\nabla F(\mathbf{X}^t)\mathbf{1}\right\|^2\right] - (-\eta/2 - \eta^2 L/2)\mathbb{E}\left[\|\bar{\mathbf{v}}^t\|^2\right] \\
&\leq F_t - \frac{\eta}{2} \mathbb{E}\left[\|\nabla f(\bar{\mathbf{x}}^t)\|^2\right] + \frac{\eta}{n^2}\hat{G}^t + \frac{\eta L^2}{n}\mathbb{E}\left[\|\mathbf{X}^t - \bar{\mathbf{x}}^t\mathbf{1}^\top\|^2\right] - (-\eta/2 - \eta^2 L/2)\Omega_5^t \\
&= F_t - \frac{\eta}{2} \mathbb{E}\left[\|\nabla f(\bar{\mathbf{x}}^t)\|^2\right] + \frac{\eta}{n^2}\hat{G}^t + \frac{\eta L^2}{n}\Omega_3^t - (-\eta/2 - \eta^2 L/2)\Omega_5^t.
\end{aligned}
$$

$\square$

**Lemma 8.** *Assume that Assumptions 2 and 3 hold. Then we have the following descent on* $\hat{G}^t$

$$
\hat{G}^{t+1} \leq (1-\lambda)\mathbb{E}\left[\|\nabla F(\mathbf{X}^t)\mathbf{1} - \mathbf{M}^t\mathbf{1}\|^2\right] + \frac{(1-\lambda)^2 n L^2}{\lambda}\mathbb{E}\left[\|\mathbf{X}^t - \mathbf{X}^{t+1}\|_{\mathrm{F}}^2\right] + \lambda^2 n \sigma^2. \quad (19)
$$

*Proof.* Using the update rules of Algorithm 1 we get

$$
\begin{aligned}
\hat{G}^{t+1} &= \mathbb{E}\left[\|\nabla F(\mathbf{X}^{t+1})\mathbf{1} - \mathbf{M}^{t+1}\mathbf{1}\|^2\right] \\
&= \mathbb{E}\left[\left\|\nabla F(\mathbf{X}^{t+1})\mathbf{1} - (1-\lambda)\mathbf{M}^t\mathbf{1} - \lambda\widetilde{\nabla}F(\mathbf{X}^{t+1})\mathbf{1}\right\|^2\right] \\
&= \mathbb{E}\left[\Big\|(1-\lambda)(\nabla F(\mathbf{X}^t) - \mathbf{M}^t)\mathbf{1} + \lambda(\nabla F(\mathbf{X}^{t+1}) - \widetilde{\nabla}F(\mathbf{X}^{t+1}))\mathbf{1}\right. \\
&\qquad\qquad \left. + (1-\lambda)(\nabla F(\mathbf{X}^{t+1}) - \nabla F(\mathbf{X}^t))\mathbf{1}\Big\|^2\right] \\
&\leq (1-\lambda)^2 \mathbb{E}\left[\|(\nabla F(\mathbf{X}^t) - \mathbf{M}^t)\mathbf{1} + (\nabla F(\mathbf{X}^{t+1}) - \nabla F(\mathbf{X}^t))\mathbf{1}\|^2\right] + \lambda^2 n \sigma^2 \\
&\leq (1-\lambda)\mathbb{E}\left[\|\nabla F(\mathbf{X}^t)\mathbf{1} - \mathbf{M}^t\mathbf{1}\|^2\right] + \frac{(1-\lambda)^2 n L^2}{\lambda}\mathbb{E}\left[\|\mathbf{X}^t - \mathbf{X}^{t+1}\|_{\mathrm{F}}^2\right] + \lambda^2 n \sigma^2,
\end{aligned}
$$

where in the first inequality we use the fact that $\mathbb{E}\left[\widetilde{\nabla}F(\mathbf{X}^{t+1})\right] = \nabla F(\mathbf{X}^{t+1})$ and Assumption 3, and in the second inequality we use $\|\mathbf{a} + \mathbf{b}\|^2 \leq (1+\beta)\|\mathbf{a}\|^2 + (1+\beta^{-1})\|\mathbf{b}\|^2$ for any vectors $\mathbf{a}, \mathbf{b}$ and constant $\mathbf{b}$. $\square$

**Lemma 9.** *Assume that Assumptions 2 and 3 hold. Then we have the following descent on* $\widetilde{G}^t$

$$
\widetilde{G}^{t+1} \leq \lambda^2 \sigma^2 n + \frac{(1-\lambda)^2 L^2}{\lambda}\mathbb{E}\left[\|\mathbf{X}^{t+1} - \mathbf{X}^t\|_{\mathrm{F}}^2\right] + (1-\lambda)\widetilde{G}^t. \quad (20)
$$

*Proof.*

$$
\begin{aligned}
\widetilde{G}^{t+1} &= \mathbb{E}\left[\left\|\nabla F(\mathbf{X}^{t+1}) - \mathbf{M}^{t+1}\right\|_{\mathrm{F}}^2\right] \\
&= \mathbb{E}\left[\left\|\nabla F(\mathbf{X}^{t+1}) - (1-\lambda)\mathbf{M}^t - \lambda\widetilde{\nabla}F(\mathbf{X}^{t+1})\right\|_{\mathrm{F}}^2\right] \\
&\leq \lambda^2\sigma^2 n + (1-\lambda)^2\mathbb{E}\left[\left\|\nabla F(\mathbf{X}^{t+1}) - \mathbf{M}^t\right\|_{\mathrm{F}}^2\right] \\
&\leq \lambda^2\sigma^2 n + (1-\lambda)^2(1+\beta_1^{-1})\mathbb{E}\left[\left\|\nabla F(\mathbf{X}^{t+1}) - \nabla F(\mathbf{X}^t)\right\|_{\mathrm{F}}^2\right] \\
&\quad + (1-\lambda)^2(1+\beta_1)\mathbb{E}\left[\left\|\mathbf{M}^t - \nabla F(\mathbf{X}^t)\right\|_{\mathrm{F}}^2\right] \\
&\leq \lambda^2\sigma^2 n + \frac{(1-\lambda)^2}{\lambda}\mathbb{E}\left[\left\|\nabla F(\mathbf{X}^{t+1}) - \nabla F(\mathbf{X}^t)\right\|_{\mathrm{F}}^2\right] + (1-\lambda)\mathbb{E}\left[\left\|\mathbf{M}^t - \nabla F(\mathbf{X}^t)\right\|_{\mathrm{F}}^2\right] \\
&\leq \lambda^2\sigma^2 n + \frac{(1-\lambda)^2 L^2}{\lambda}\mathbb{E}\left[\left\|\mathbf{X}^{t+1} - \mathbf{X}^t\right\|_{\mathrm{F}}^2\right] + (1-\lambda)\widetilde{G}^t.
\end{aligned}
$$

where we choose $\beta_1 = \frac{\lambda}{(1-\lambda)}$. $\qquad\square$

**Lemma 10.** *Let $\mathcal{C}_\alpha$ be any contractive compressor with parameter $\alpha$. Then we have the following descent on $\Omega_1^t$*

$$
\Omega_1^{t+1} \leq (1-\alpha/2)\mathbb{E}\left[\left\|\mathbf{H}^t - \mathbf{X}^t\right\|_{\mathrm{F}}^2\right] + \frac{2}{\alpha}\mathbb{E}\left[\left\|\mathbf{X}^t - \mathbf{X}^{t+1}\right\|_{\mathrm{F}}^2\right]. \tag{21}
$$

*Proof.* We have

$$
\begin{aligned}
\Omega_1^{t+1} &= \mathbb{E}\left[\left\|\mathbf{H}^{t+1} - \mathbf{X}^{t+1}\right\|_{\mathrm{F}}^2\right] \\
&= \mathbb{E}\left[\left\|\mathbf{H}^t + \mathcal{C}_\alpha(\mathbf{X}^{t+1} - \mathbf{H}^t) - \mathbf{X}^{t+1}\right\|_{\mathrm{F}}^2\right] \\
&\leq (1-\alpha)\mathbb{E}\left[\left\|\mathbf{H}^t - \mathbf{X}^{t+1}\right\|_{\mathrm{F}}^2\right] \\
&\leq (1-\alpha/2)\mathbb{E}\left[\left\|\mathbf{H}^t - \mathbf{X}^t\right\|_{\mathrm{F}}^2\right] + \frac{2}{\alpha}\mathbb{E}\left[\left\|\mathbf{X}^t - \mathbf{X}^{t+1}\right\|_{\mathrm{F}}^2\right].
\end{aligned}
$$

$\qquad\square$

**Lemma 11.** *Let $\mathcal{C}_\alpha$ be any contractive compressor with parameter $\alpha$. Then we have the following descent on $\Omega_2^t$*

*Proof.* The proof is similar to the one of Lemma 10

$$
\begin{aligned}
\Omega_2^{t+1} &= \mathbb{E}\left[\left\|\mathbf{G}^{t+1} - \mathbf{V}^{t+1}\right\|_{\mathrm{F}}^2\right] \\
&\leq (1-\alpha/2)\mathbb{E}\left[\left\|\mathbf{G}^t - \mathbf{V}^t\right\|_{\mathrm{F}}^2\right] + \frac{2}{\alpha}\mathbb{E}\left[\left\|\mathbf{V}^t - \mathbf{V}^{t+1}\right\|_{\mathrm{F}}^2\right].
\end{aligned}
$$

$\qquad\square$

**Lemma 12.** *We have the following descent on $\Omega_3^t$*

$$
\Omega_3^{t+1} \leq (1-\gamma\rho/2)\Omega_3^t + (1+2/\gamma\rho)2\gamma^2 C\Omega_1^t + (1+2/\gamma\rho)2\eta^2\Omega_4^t. \tag{22}
$$

*Proof.*

$$\Omega_3^{t+1} = \mathbb{E}\left[\left\|\mathbf{X}^{t+1} - \bar{\mathbf{x}}^{t+1}\mathbf{1}^\top\right\|_{\mathrm{F}}^2\right]$$

$$= \mathbb{E}\left[\left\|\mathbf{X}^t + \gamma\mathbf{H}^t(\mathbf{W} - \mathbf{I}) - \eta\mathbf{V}^t - \bar{\mathbf{x}}^t\mathbf{1}^T + \eta\bar{\mathbf{v}}^t\mathbf{1}^T\right\|_{\mathrm{F}}^2\right]$$

$$= \mathbb{E}\left[\left\|\mathbf{X}^t\widetilde{\mathbf{W}} - \bar{\mathbf{x}}^t\mathbf{1}^T + \gamma(\mathbf{H}^t - \mathbf{X}^t)(\mathbf{W} - \mathbf{I}) - \eta\mathbf{V}^t + \eta\bar{\mathbf{v}}^t\mathbf{1}^\top\right\|_{\mathrm{F}}^2\right]$$

$$\leq (1 + \beta)(1 - \gamma\rho)\mathbb{E}\left[\left\|\mathbf{X}^t - \bar{\mathbf{x}}^t\mathbf{1}^\top\right\|_{\mathrm{F}}^2\right] + (1 + \beta^{-1})(2\gamma^2\mathbb{E}\left[\left\|(\mathbf{H}^t - \mathbf{X}^t)(\mathbf{W} - \mathbf{I})\right\|_{\mathrm{F}}^2\right]$$

$$+ 2\eta^2\mathbb{E}\left[\left\|\mathbf{V}^t - \bar{\mathbf{v}}^t\mathbf{1}^\top\right\|_{\mathrm{F}}^2\right])$$

$$\leq (1 - \gamma\rho/2)\mathbb{E}\left[\left\|\mathbf{X}^t - \bar{\mathbf{x}}^t\mathbf{1}^\top\right\|_{\mathrm{F}}^2\right] + (1 + 2/\gamma\rho)(2\gamma^2 C\mathbb{E}\left[\left\|\mathbf{H}^t - \mathbf{X}^t\right\|_{\mathrm{F}}^2\right]$$

$$+ 2\eta^2\mathbb{E}\left[\left\|\mathbf{V}^t - \bar{\mathbf{v}}^t\mathbf{1}^\top\right\|_{\mathrm{F}}^2\right])$$

$$= (1 - \gamma\rho/2)\Omega_3^t + (1 + 2/\gamma\rho)2\gamma^2 C\Omega_1^t + (1 + 2/\gamma\rho)2\eta^2\Omega_4^t.$$

where $\beta = \frac{\gamma\rho/2}{1 - \gamma\rho}$ and we define $\widetilde{\mathbf{W}} := \mathbf{I} + \gamma(\mathbf{W} - \mathbf{I})$ which has a spectral gap at least $\gamma\rho$ by Lemma 5. $\qquad\square$

**Lemma 13.** *We have the following descent on $\Omega_4^t$*

$$\Omega_4^{t+1} \leq (1 - \gamma\rho/2)\mathbb{E}\left[\left\|\mathbf{V}^t - \bar{\mathbf{v}}^t\mathbf{1}^T\right\|_{\mathrm{F}}^2\right]$$

$$+ (1 + 2/\gamma\rho)\left(2\gamma^2 C\mathbb{E}\left[\left\|\mathbf{G}^t - \mathbf{V}^t\right\|_{\mathrm{F}}^2\right] + 2\mathbb{E}\left[\left\|\mathbf{M}^{t+1} - \mathbf{M}^t\right\|_{\mathrm{F}}^2\right]\right). \qquad (23)$$

*Proof.*

$$\Omega_4^{t+1} = \mathbb{E}\left[\left\|\mathbf{V}^{t+1} - \bar{\mathbf{v}}^t\mathbf{1}^T + \bar{\mathbf{v}}^t\mathbf{1}^T - \bar{\mathbf{v}}^{t+1}\mathbf{1}^T\right\|_{\mathrm{F}}^2\right]$$

$$= \mathbb{E}\left[\left\|\mathbf{V}^{t+1} - \bar{\mathbf{v}}^t\mathbf{1}^T\right\|_{\mathrm{F}}^2\right] - n\mathbb{E}\left[\left\|\bar{\mathbf{v}}^t - \bar{\mathbf{v}}^{t+1}\right\|^2\right]$$

$$\leq \mathbb{E}\left[\left\|\mathbf{V}^{t+1} - \bar{\mathbf{v}}^t\mathbf{1}^T\right\|_{\mathrm{F}}^2\right]$$

$$= \mathbb{E}\left[\left\|\mathbf{V}^t + \gamma\mathbf{G}^t(\mathbf{W} - \mathbf{I}) + \mathbf{M}^{t+1} - \mathbf{M}^t - \bar{\mathbf{v}}^t\mathbf{1}^T\right\|_{\mathrm{F}}^2\right]$$

$$= \mathbb{E}\left[\left\|\mathbf{V}^t\widetilde{\mathbf{W}} - \bar{\mathbf{v}}^t\mathbf{1}^T + \gamma(\mathbf{G}^t - \mathbf{V}^t)(\mathbf{W} - \mathbf{I}) + \mathbf{M}^{t+1} - \mathbf{M}^t\right\|_{\mathrm{F}}^2\right]$$

$$\leq (1 - \gamma\rho/2)\mathbb{E}\left[\left\|\mathbf{V}^t - \bar{\mathbf{v}}^t\mathbf{1}^T\right\|_{\mathrm{F}}^2\right] + (1 + 2/\gamma\rho)(2\gamma^2 C\mathbb{E}\left[\left\|\mathbf{G}^t - \mathbf{V}^t\right\|_{\mathrm{F}}^2\right]$$

$$+ 2\mathbb{E}\left[\left\|\mathbf{M}^{t+1} - \mathbf{M}^t\right\|_{\mathrm{F}}^2\right]).$$

$$\square$$

**Lemma 14** (Lemma B.4, Eq. (18) from (Zhao et al., 2022)). *We have the following control of the iterates at iterations $t$ and $t + 1$*

$$\mathbb{E}\left[\left\|\mathbf{X}^{t+1} - \mathbf{X}^t\right\|_{\mathrm{F}}^2\right] \leq 3\gamma^2 C\Omega_1^t + 3\gamma^2 C\Omega_3^t + 3\eta^2\Omega_4^t + 3\eta^2 n\Omega_5^t. \qquad (24)$$

**Lemma 15.** *Assume Assumptions 2 and 3 hold. Then we have the following control of the momentum at iterations $t$ and $t + 1$*

$$\mathbb{E}\left[\left\|\mathbf{M}^{t+1} - \mathbf{M}^t\right\|_{\mathrm{F}}^2\right] \leq \lambda^2 n\sigma^2 + 2\lambda^2\mathbb{E}\left[\left\|\nabla F(\mathbf{X}^t) - \mathbf{M}^t\right\|_{\mathrm{F}}^2\right] + 2\lambda^2 L^2\mathbb{E}\left[\left\|\mathbf{X}^t - \mathbf{X}^{t+1}\right\|_{\mathrm{F}}^2\right]. \qquad (25)$$

*Proof.*

$$\mathbb{E}\left[\left\|\mathbf{M}^{t+1} - \mathbf{M}^t\right\|_{\mathrm{F}}^2\right] = \lambda^2 \mathbb{E}\left[\left\|\widetilde{\nabla} F(\mathbf{X}^{t+1}) - \mathbf{M}^t\right\|_{\mathrm{F}}^2\right]$$

$$= \lambda^2 \mathbb{E}\left[\left\|\widetilde{\nabla} F(\mathbf{X}^{t+1}) - \nabla F(\mathbf{X}^{t+1}) + \nabla F(\mathbf{X}^{t+1}) - \mathbf{M}^t\right\|_{\mathrm{F}}^2\right]$$

$$\leq \lambda^2 n\sigma^2 + \lambda^2 \mathbb{E}\left[\left\|\nabla F(\mathbf{X}^{t+1}) - \mathbf{M}^t\right\|_{\mathrm{F}}^2\right]$$

$$\leq \lambda^2 n\sigma^2 + 2\lambda^2 \mathbb{E}\left[\left\|\nabla F(\mathbf{X}^t) - \mathbf{M}^t\right\|_{\mathrm{F}}^2\right] + 2\lambda^2 L^2 \mathbb{E}\left[\left\|\mathbf{X}^t - \mathbf{X}^{t+1}\right\|_{\mathrm{F}}^2\right].$$

$\square$

**Lemma 16.** *We have the following control of the gradient estimator $\mathbf{V}^t$ at iterations $t$ and $t+1$*

$$\mathbb{E}\left[\left\|\mathbf{V}^{t+1} - \mathbf{V}^t\right\|_{\mathrm{F}}^2\right] \leq 3\gamma^2 C\Omega_2^t + 3\gamma^2 C\Omega_4^t + 3\mathbb{E}\left[\left\|\mathbf{M}^{t+1} - \mathbf{M}^t\right\|_{\mathrm{F}}^2\right]. \tag{26}$$

*Proof.*

$$\mathbb{E}\left[\left\|\mathbf{V}^{t+1} - \mathbf{V}^t\right\|_{\mathrm{F}}^2\right] = \mathbb{E}\left[\left\|\gamma\mathbf{G}^t(\mathbf{W} - \mathbf{I}) + \mathbf{M}^{t+1} - \mathbf{M}^t\right\|_{\mathrm{F}}^2\right]$$

$$= \mathbb{E}\left[\left\|\gamma(\mathbf{G}^t - \mathbf{V}^t)(\mathbf{W} - \mathbf{I}) + \gamma(\mathbf{V}^t - \bar{\mathbf{v}}^t\mathbf{1}^T)(\mathbf{W} - \mathbf{I}) + \mathbf{M}^{t+1} - \mathbf{M}^t\right\|_{\mathrm{F}}^2\right]$$

$$\leq 3\gamma^2 C\mathbb{E}\left[\left\|\mathbf{G}^t - \mathbf{V}^t\right\|_{\mathrm{F}}^2\right] + 3\gamma^2 C\mathbb{E}\left[\left\|\mathbf{V}^t - \bar{\mathbf{v}}^t\mathbf{1}^T\right\|_{\mathrm{F}}^2\right]$$

$$+ 3\mathbb{E}\left[\left\|\mathbf{M}^{t+1} - \mathbf{M}^t\right\|_{\mathrm{F}}^2\right]$$

$$= 3\gamma^2 C\Omega_2^t + 3\gamma^2 C\Omega_4^t + 3\mathbb{E}\left[\left\|\mathbf{M}^{t+1} - \mathbf{M}^t\right\|_{\mathrm{F}}^2\right].$$

$\square$

**Theorem 1** (Convergence of MoTEF). *Let Assumptions 2 and 3 hold. Then there exist absolute constants $c_\gamma, c_\lambda, c_\eta$, and some $\tau \leq 1$ such that if we set stepsizes $\gamma = c_\gamma \alpha\rho, \lambda = c_\lambda \alpha\rho^3\tau, \eta = c_\eta L^{-1}\alpha\rho^3\tau$, and choosing the initial batch size $B_{\mathrm{init}} \geq \lceil \frac{LF^0}{\sigma^2} \rceil$, then after at most*

$$T = \mathcal{O}\left(\frac{\sigma^2}{n\varepsilon^4} + \frac{\sigma}{\alpha\rho^{5/2}\varepsilon^3} + \frac{1}{\alpha\rho^3\varepsilon^2}\right) LF^0 \tag{10}$$

*iterations of Algorithm 1 it holds $\mathbb{E}\left[\|\nabla f(\mathbf{x}_{\mathrm{out}})\|^2\right] \leq \varepsilon^2$, where $\mathbf{x}_{\mathrm{out}}$ is chosen uniformly at random from $\{\bar{\mathbf{x}}_0, \ldots, \bar{\mathbf{x}}_{T-1}\}$, and $\mathcal{O}$ suppresses absolute constants.*

*Proof.* From Lemma 15 and Lemma 14 we get

$$\mathbb{E}\left[\left\|\mathbf{M}^{t+1} - \mathbf{M}^t\right\|_{\mathrm{F}}^2\right] \leq \lambda^2 n\sigma^2 + 2\lambda^2 \widetilde{G}^t + 6\lambda^2\gamma^2 L^2 C\Omega_1^t + 6\lambda^2\gamma^2 L^2 C\Omega_3^t + 6\lambda^2\eta^2 L^2\Omega_4^t$$

$$+ 6\lambda^2\eta^2 L^2 n\Omega_5^t. \tag{27}$$

Using the above and Lemma 16 we get

$$\mathbb{E}\left[\left\|\mathbf{V}^{t+1} - \mathbf{V}^t\right\|_{\mathrm{F}}^2\right] \leq 3\lambda^2 n\sigma^2 + 6\lambda^2 \widetilde{G}^t + 18\lambda^2\gamma^2 L^2 C\Omega_1^t + 3\gamma^2 C\Omega_2^t + 18\lambda^2\gamma^2 L^2 C\Omega_3^t$$

$$+ (3\gamma^2 C + 18\lambda^2\eta^2 L^2)\Omega_4^t + 18\lambda^2\eta^2 L^2 n\Omega_5^t. \tag{28}$$

Using (27), (28), and Lemma 8 we get the following descent on $\hat{G}^t$

$$\hat{G}^{t+1} \leq (1 - \lambda)\hat{G}^t + \frac{3L^2 n\gamma^2 C}{\lambda}\Omega_1^t + \frac{3L^2 n\gamma^2 C}{\lambda}\Omega_3^t + \frac{3L^2 n\eta^2}{\lambda}\Omega_4^t + \frac{3L^2 n^2\eta^2}{\lambda}\Omega_5^t + \lambda^2 n\sigma^2.$$

Using (27), (28), and Lemma 9 we get the following descent on $\widetilde{G}^t$

$$\widetilde{G}^{t+1} \leq (1 - \lambda)\widetilde{G}^t + \frac{3L^2\gamma^2 C}{\lambda}\Omega_1 + \frac{3L^2\gamma^2 C}{\lambda}\Omega_3^t + \frac{3L^2\eta^2}{\lambda}\Omega_4^t + \frac{3L^2 n\eta^2}{\lambda}\Omega_5^t + \lambda^2 n\sigma^2.$$

Using (27), (28), and Lemma 10 we get the following descent on $\Omega_1^t$

$$\Omega_1^{t+1} \leq \left(1 - \frac{\alpha}{2} + \frac{6\gamma^2 C}{\alpha}\right)\Omega_1^t + \frac{6\gamma^2 C}{\alpha}\Omega_3^t + \frac{6\eta^2}{\alpha}\Omega_4^t + \frac{6\eta^2 n}{\alpha}\Omega_5^t.$$

Using (27), (28), and Lemma 11 we get the following descent on $\Omega_2^t$

$$\Omega_2^{t+1} \leq \left(1 - \frac{\alpha}{2} + \frac{6\gamma^2 C}{\alpha}\right)\Omega_2^t + \frac{6\lambda^2}{\alpha}\widetilde{G}^t + \frac{36\lambda^2\gamma^2 L^2 C}{\alpha}\Omega_1^t + \frac{36\lambda^2\gamma^2 L^2 C}{\alpha}\Omega_3^t$$

$$+ \left(\frac{6\gamma^2 C}{\alpha} + \frac{36\eta^2\lambda^2 L^2}{\alpha}\right)\Omega_4^t + \frac{36\eta^2\lambda^2 L^2 n}{\alpha}\Omega_5^t + \frac{6\lambda^2 n}{\alpha}\sigma^2.$$

Using (27), (28), and Lemma 12 we get the following descent on $\Omega_3^t$

$$\Omega_3^{t+1} \leq (1 - \frac{\gamma\rho}{2})\Omega_3^t + \frac{6\gamma C}{\rho}\Omega_1^t + \frac{6\eta^2}{\gamma\rho}\Omega_4^t. \tag{29}$$

Finally, using (27), (28), and Lemma 13 we get the following descent on $\Omega_4^t$:

$$\Omega_4^{t+1} \leq (1 - \frac{\gamma\rho}{2} + \frac{36\eta^2\lambda^2 L^2}{\gamma\rho})\Omega_4^t + \frac{12\lambda^2}{\gamma\rho}\widetilde{G}^t + \frac{36\gamma\lambda^2 L^2 C}{\rho}\Omega_1^t + \frac{6\gamma C}{\rho}\Omega_2^t + \frac{36\gamma\lambda^2 L^2 C}{\rho}\Omega_3^t$$

$$+ \frac{36\eta^2\gamma L^2 n}{\rho}\Omega_5^t + \frac{6\lambda^2 n}{\gamma\rho}\sigma^2.$$

Now we can gather all together

$$\mathbf{\Omega}^{t+1} \leq \underbrace{\begin{pmatrix} 1-\lambda & 0 & \frac{3L^2 n\gamma^2 C}{\lambda} & 0 & \frac{3L^2 n\gamma^2 C}{\lambda} & \frac{3L^2 n\eta^2}{\lambda} \\ 0 & 1-\lambda & \frac{3L^2\gamma^2 C}{\lambda} & 0 & \frac{3L^2\gamma^2 C}{\lambda} & \frac{3L^2\eta^2}{\lambda} \\ 0 & 0 & 1-\frac{\alpha}{2}+\frac{6\gamma^2 C}{\alpha} & 0 & \frac{6\gamma^2 C}{\alpha} & \frac{6\eta^2}{\alpha} \\ 0 & \frac{6\lambda^2}{\alpha} & \frac{36\lambda^2\gamma^2 L^2 C}{\alpha} & 1-\frac{\alpha}{2}+\frac{6\gamma^2 C}{\alpha} & \frac{36\lambda^2\gamma^2 L^2 C}{\alpha} & \frac{6\gamma^2 C}{\alpha}+\frac{36\lambda^2\eta^2 L^2}{\alpha} \\ 0 & 0 & \frac{6\gamma C}{\rho} & 0 & 1-\frac{\gamma\rho}{2} & \frac{6\eta^2}{\gamma\rho} \\ 0 & \frac{12\lambda^2}{\gamma\rho} & \frac{36\gamma\lambda^2 L^2 C}{\rho} & \frac{6\gamma C}{\rho} & \frac{36\gamma\lambda^2 L^2 C}{\rho} & 1-\frac{\gamma\rho}{2}+\frac{36\eta^2\lambda^2 L^2}{\gamma\rho} \end{pmatrix}}_{:=\mathbf{A}} \mathbf{\Omega}^t$$

$$+ \underbrace{\begin{pmatrix} \frac{3L^2 n^2\eta^2}{\lambda} \\ \frac{3L^2 n\eta^2}{\lambda} \\ \frac{6\eta^2 n}{\alpha} \\ \frac{36\eta^2\lambda^2 L^2 n}{\alpha} \\ 0 \\ \frac{36\eta^2\gamma L^2 n}{\rho} \end{pmatrix}}_{:=\mathbf{b}_1} \Omega_5^t + \underbrace{\begin{pmatrix} n \\ 2n \\ 0 \\ \frac{6n}{\alpha} \\ 0 \\ \frac{6n}{\gamma\rho} \end{pmatrix}}_{:=\mathbf{b}_2} \lambda^2\sigma^2. \tag{30}$$

We remind that the Lyapunov function $\Phi^t$ has the following form

$$\Phi^t := F^t + \frac{c_1}{n^2 L}\hat{G}^t + \frac{c_2\tau}{nL}\widetilde{G}^t + \frac{c_3 L}{\rho^3 n\tau}\Omega_1^t + \frac{c_4\tau}{\rho nL}\Omega_2^t + \frac{c_5 L}{\rho^3 n\tau}\Omega_3^t + \frac{c_6\tau}{\rho nL}\Omega_4^t = F^t + \mathbf{c}^\top\mathbf{\Omega}^t,$$

where $\{c_k\}_{k=1}^6$ are absolute constants. Let

$$\mathbf{c} := \left(\frac{c_1}{n^2 L}, \frac{c_2\tau}{nL}, \frac{c_3 L}{\rho^3 n\tau}, \frac{c_4\tau}{\rho nL}, \frac{c_5 L}{\rho^3 n\tau}, \frac{c_6\tau}{\rho nL}\right)^\top.$$

Therefore, the descent on $\Phi^t$ for is the following

$$\Phi^{t+1} = F^{t+1} + \mathbf{c}^\top \mathbf{\Omega}^t$$

$$\leq F_t - \frac{\eta}{2}\mathbb{E}\left[\left\|\nabla f(\bar{\mathbf{x}}^t)\right\|^2\right] + \frac{\eta}{n^2}\hat{G}^t + \frac{\eta L^2}{n}\Omega_3^t - (\eta/2 - \eta^2 L/2)\Omega_5^t$$

$$+ \mathbf{c}^\top(\mathbf{A}\mathbf{\Omega}^t + \Omega_5^t\mathbf{b}_1 + \lambda^2\sigma^2\mathbf{b}_2)$$

$$= F^t - \frac{\eta}{2}\mathbb{E}\left[\left\|\nabla f(\bar{\mathbf{x}}^t)\right\|^2\right] + \mathbf{c}^\top\mathbf{\Omega}^t + (\mathbf{q}^\top + \mathbf{c}^\top\mathbf{A} - \mathbf{c}^\top)\mathbf{\Omega}^t - (\eta/2 - \eta^2 L/2 - \mathbf{c}^\top\mathbf{b}_1)\Omega_5^t$$

$$+ \mathbf{c}^\top\mathbf{b}_2\lambda^2\sigma^2$$

$$= \Phi^t - \frac{\eta}{2}\mathbb{E}\left[\left\|\nabla f(\bar{\mathbf{x}}^t)\right\|^2\right] + (\mathbf{q}^\top + \mathbf{c}^\top\mathbf{A} - \mathbf{c}^\top)\mathbf{\Omega}^t - (\eta/2 - \eta^2 L/2 - \mathbf{c}^\top\mathbf{b}_1)\Omega_5^t$$

$$+ \mathbf{c}^\top\mathbf{b}_2\lambda^2\sigma^2,$$

where $\mathbf{q} := (\eta/n^2, 0, 0, 0, \eta L^2/n, 0)^\top$. We need coefficients next to $\mathbf{\Omega}^t$ and $\Omega_5^t$ to be negative. This is equivalent to finding $\mathbf{c}$ such that

$$\begin{bmatrix} \mathbf{I} - \mathbf{A}^\top \\ -\mathbf{b}_1^\top \end{bmatrix} \mathbf{c} \geq \begin{bmatrix} \mathbf{q} \\ \frac{\eta^2 L}{2} - \frac{\eta}{2} \end{bmatrix}. \tag{31}$$

We make the following choice of stepsizes

$$\lambda := c_\lambda \alpha \rho^3 \tau, \quad \gamma := c_\gamma \alpha \rho, \quad \eta := \frac{c_\eta \alpha \rho^3 \tau}{L}.$$

with the following choice of constants:

$$c_\lambda = \frac{1}{200}, c_\gamma = \frac{1}{200}, c_\eta = \frac{1}{100000}, \quad \text{and,}$$

$$c_1 = \frac{1}{500}, c_2 = \frac{13}{200000}, c_3 = \frac{1}{20}, c_4 = \frac{1}{400000}, c_5 = \frac{9}{100}, c_6 = \frac{1}{200000}. \tag{32}$$

The system of inequalities (31) are satisfied when $\tau \leq 1$.

Given the complexity of the inequalities and the choices of the parameters, we do not attempt to write down a proof for the correctness of the choices manually, instead, we verify these choices using the Symbolic Math Toolbox in MATLAB. We also perform such verification for our parameters and constants choices for MoTEF in PŁ case and MoTEF-VR. The code performing all the verification can be found at this anonymous link. We also note that, when $c_\lambda$, $c_\gamma$ and $c_\eta$ are fixed, we can search for a feasible $\{c_i\}_{i\in[6]}$ efficiently using the Linear Program solver with MATLAB as well. But searching for a feasible set of choices for $c_\lambda$, $c_\gamma$ and $c_\eta$ is very much a trial-and-error process.

Note that this choice gives us both $\lambda$ and $\gamma$ smaller than 1. This choice of constants gives the following result

$$\begin{aligned}
\Phi^{t+1} &\leq \Phi^t - \frac{c_\eta \alpha \rho^3 \tau}{2L}\mathbb{E}\left[\left\|\nabla f(\bar{\mathbf{x}}^t)\right\|^2\right] + \frac{c_1}{n^2 L} \cdot nc_\lambda^2\alpha^2\rho^6\tau^2\sigma^2 \\
&\quad + \frac{c_2\tau}{nL} \cdot 2nc_\lambda^2\alpha^2\rho^6\tau^2\sigma^2 \\
&\quad + \frac{c_4\tau}{\rho nL} \cdot \frac{6n}{\alpha}c_\lambda^2\alpha^2\rho^6\tau^2\sigma^2 \\
&\quad + \frac{c_6\tau}{\rho nL} \cdot \frac{6n}{c_\gamma\alpha\rho}c_\lambda^2\alpha^2\rho^6\tau^2\sigma^2 \\
&= \Phi^t - \frac{c_\eta\alpha\rho^3\tau}{2L}\mathbb{E}\left[\left\|\nabla f(\bar{\mathbf{x}}^t)\right\|^2\right] + \frac{c_\lambda^2 c_1\alpha^2\rho^6}{nL} \cdot \tau^2\sigma^2 \\
&\quad + \left(\frac{6c_\lambda^2 c_4\alpha\rho^5}{L} + \frac{2c_\lambda^2 c_2\alpha^2\rho^6}{L} + \frac{6c_6 c_\lambda^2\alpha\rho^4}{c_\gamma L}\right)\tau^3\sigma^2. \tag{33}
\end{aligned}$$

By this, we proved Lemma 1. Let us define constants

$$
\begin{aligned}
B &:= \frac{c_\eta \alpha \rho^3}{2L}, \\
C &:= \frac{c_\lambda^2 c_1 \alpha^2 \rho^6}{nL}, \\
D &:= \left( \frac{6 c_\lambda^2 c_4 \alpha \rho^5}{L} + \frac{2 c_\lambda^2 c_2 \alpha^2 \rho^6}{L} + \frac{6 c_6 c_\lambda^2 \alpha \rho^4}{c_\gamma L} \right), \\
E &:= 1.
\end{aligned}
$$

Using $\tau \le E$ and unrolling (33) for $T$ iterations we get

$$
\frac{1}{T} \sum_{t=0}^{T-1} \mathbb{E}\left[\|\nabla f(\bar{\mathbf{x}}^t)\|^2\right] \le \frac{\Phi^0}{\tau BT} + \frac{C}{B}\tau\sigma^2 + \frac{D}{B}\tau^2\sigma^2.
$$

So we need to choose $\tau = \min\left\{ \frac{1}{E}, \left(\frac{\Phi^0}{CT\sigma^2}\right)^{1/2}, \left(\frac{\Phi^0}{DT\sigma^2}\right)^{1/3} \right\}$ and we get the following rate

$$
\begin{aligned}
\frac{1}{T} \sum_{t=0}^{T-1} \mathbb{E}\left[\|\nabla f(\bar{\mathbf{x}}^t)\|^2\right] &\le \mathcal{O}\left( \frac{\Phi^0 E}{BT} + \left(\frac{C\Phi^0\sigma^2}{B^2T}\right)^{1/2} + \left(\frac{\sqrt{D}\Phi^0\sigma}{B^{3/2}T}\right)^{2/3} \right) \\
&= \mathcal{O}\left( \frac{L\Phi^0}{\alpha\rho^3 T} + \left(\frac{L\Phi^0\sigma^2}{nT}\right)^{1/2} \right. \\
&\quad + \left. \left(\frac{\sqrt{\alpha\rho^5 + \alpha^2\rho^6 + \alpha\rho^4}L\Phi^0\sigma}{\alpha^{3/2}\rho^{9/2}T}\right)^{2/3} \right) \\
&= \mathcal{O}\left( \frac{L\Phi^0}{\alpha\rho^3 T} + \left(\frac{L\Phi^0\sigma^2}{nT}\right)^{1/2} + \left(\frac{(\rho^{1/2}+\alpha^{1/2}\rho+1)L\Phi^0\sigma}{\alpha\rho^{5/2}T}\right)^{2/3} \right),
\end{aligned}
$$

that translates to the rate in terms of $\varepsilon$ to

$$
T = \mathcal{O}\left( \frac{L\Phi^0}{\alpha\rho^3\varepsilon^2} + \frac{L\Phi^0\sigma^2}{n\varepsilon^4} + \frac{L\Phi^0\sigma}{\alpha\rho^2\varepsilon^3} + \frac{L\Phi^0\sigma}{\alpha^{1/2}\rho^{3/2}\varepsilon^3} + \frac{L\Phi^0\sigma}{\alpha\rho^{5/2}\varepsilon^3} \right) \Rightarrow \frac{1}{T}\sum_{t=0}^{T-1}\mathbb{E}\left[\|\nabla f(\bar{\mathbf{x}}^t)\|^2\right] \le \varepsilon^2.
$$

In the result above the fifth term always dominates the third and fourth. Therefore, we remove the third and fourth terms from the rate and derive the following rate

$$
T = \mathcal{O}\left( \frac{L\Phi^0}{\alpha\rho^3\varepsilon^2} + \frac{L\Phi^0\sigma^2}{n\varepsilon^4} + \frac{L\Phi^0\sigma}{\alpha\rho^{5/2}\varepsilon^3} \right) \Rightarrow \frac{1}{T}\sum_{t=0}^{T-1}\mathbb{E}\left[\|\nabla f(\bar{\mathbf{x}}^t)\|^2\right] \le \varepsilon^2.
$$

Note that with the choice $\mathbf{V}^0 = \mathbf{G}^0 = \mathbf{M}^0 = \widetilde{\nabla} F(\mathbf{X}^0), \mathbf{H}^0 = \mathbf{X}^0 = \mathbf{x}^0\mathbf{1}^\top$, we get

$$
\hat{G}^0 \le \sigma^2 n, \quad \widetilde{G}^0 \le \sigma^2 n, \quad \Omega_1^0 = \Omega_2^0 = \Omega_3^0 = \Omega_4^0 = 0.
$$

$$
\Phi^0 \le F^0 + \frac{c_1}{n^2 L}\sigma^2 n + \frac{c_2\tau}{nL}\sigma^2 n. \tag{34}
$$

If we choose the initial batch size $B_{\text{init}} \ge \lceil \frac{\sigma^2}{LF^0} \rceil$, we get

$$
\Phi^0 \le F^0 + \frac{1}{nL}\frac{\sigma^2}{B_{\text{init}}} + \frac{1}{L}\frac{\sigma^2}{B_{\text{init}}} \le 3F^0. \tag{35}
$$

$\square$

### B.1.1 Convergence of Consensus Error

Now we show that the workers achieve consensus automatically with MoTEF. We notice that (33) can be tighten. In particular, if we substitute the choices of constants in $\mathbf{c}$ into (31), we have the following:

$$(\mathbf{q}^\top + \mathbf{c}^\top \mathbf{A} - \mathbf{c}^\top)\boldsymbol{\Omega}^t \le -c_7 \frac{L\alpha}{\rho\tau n}\Omega_3^t$$

where $c_7$ is an absolute constant. We highlight that the choice of constants $\{c_k\}_{k=1}^7$ can be tightened but we are interested in the dependency on the problem-specific parameters only. In particular, this implies that we have the following (instead of (33)):

$$
\begin{aligned}
\Phi^{t+1} &= \Phi^t - \frac{c_\eta \alpha \rho^3 \tau}{2L}\mathbb{E}\left[\|\nabla f(\bar{\mathbf{x}}^t)\|^2\right] - c_7 \frac{L\alpha}{\rho\tau n}\Omega_3^t + \frac{c_\lambda^2 c_1 \alpha^2 \rho^6}{nL}\cdot\tau^2\sigma^2 \\
&+ \left(\frac{6c_\lambda^2 c_4 \alpha \rho^5}{L} + \frac{2c_\lambda^2 c_2 \alpha^2 \rho^6}{L} + \frac{6c_6 c_\lambda^2 \alpha \rho^4}{c_\gamma L}\right)\tau^3\sigma^2.
\end{aligned}
\tag{36}
$$

Therefore, we have:

$$\frac{1}{T}\sum_{t=0}^{T-1}\mathbb{E}\left[\|\nabla f(\bar{\mathbf{x}}^t)\|^2\right] + \frac{2c_7 L^2}{c_\eta \rho^4 \tau^2}\frac{1}{T}\sum_{t=0}^{T-1}\frac{1}{n}\Omega_3^t \le \frac{\Phi^0}{\tau BT} + \frac{C}{B}\tau\sigma^2 + \frac{D}{B}\tau^2\sigma^2.$$

where $B, C, D$ are defined in the proof of Theorem 1 as before. In particular, this means that $\frac{2c_7 L^2}{c_\eta \rho^4 \tau^2}\frac{1}{T}\sum_{t=0}^{T-1}\frac{1}{n}\Omega_3^t$ converges to zero at the same speed as $\frac{1}{T}\sum_{t=0}^{T-1}\mathbb{E}\left[\|\nabla f(\bar{\mathbf{x}}^t)\|^2\right]$. By our choice of $\tau \le 1$, we have:

$$
\begin{aligned}
\frac{1}{Tn}\sum_{t=0}^{T-1}\Omega_3^t &\le \frac{c_\eta \rho^4}{2c_7 L^2}\left(\frac{\Phi^0}{\tau BT} + \frac{C}{B}\tau\sigma^2 + \frac{D}{B}\tau^2\sigma^2\right) \\
&\le \mathcal{O}\left(\frac{\rho\Phi^0}{\alpha LT} + \left(\frac{\rho^8 \Phi^0 \sigma^2}{nL^3 T}\right)^{1/2} + \left(\frac{(\rho^4 + \alpha^{1/2}\rho^{7/2} + \rho^{7/2})\Phi^0\sigma}{\alpha L^2 T}\right)^{2/3}\right).
\end{aligned}
$$

Therefore, we obtain that

$$T = \mathcal{O}\left(\frac{\rho\Phi^0}{\alpha L\varepsilon^2} + \frac{\rho^8 \Phi^0 \sigma^2}{nL^3 \varepsilon^4} + \frac{\rho^{7/2}L\Phi^0\sigma}{\alpha L^2 \varepsilon^3}\right) \Rightarrow \frac{1}{Tn}\sum_{t=0}^{T-1}\Omega_3^t \le \varepsilon^2.$$

### B.1.2 Convergence of Local Models

Since we have the convergence of the averaged gradient norm $\frac{1}{T}\sum_{t=0}^{T-1}\mathbb{E}\|\nabla f(\bar{\mathbf{x}}^t)\|^2$ and the consensus error $\frac{1}{Tn}\sum_{t=0}^{T-1}\Omega_3^t$, we also obtain the convergence of local models. Indeed, we have

$$
\begin{aligned}
\frac{1}{Tn}\sum_{t=0}^{T-1}\sum_{i=1}^{n}\mathbb{E}\left[\|\nabla f(\mathbf{x}_i^t)\|^2\right] &\le \frac{2}{Tn}\sum_{t=0}^{T-1}\sum_{i=1}^{n}\mathbb{E}\left[\|\nabla f(\bar{\mathbf{x}}^t) - \nabla f(\mathbf{x}_i^t)\|^2\right] \\
&+ \frac{2}{Tn}\sum_{t=0}^{T-1}\sum_{i=1}^{n}\mathbb{E}\left[\|\nabla f(\bar{\mathbf{x}}^t)^2\|^2\right] \\
&\le \frac{2L^2}{Tn}\sum_{t=0}^{T-1}\mathbb{E}\left[\|\bar{\mathbf{x}}^t - \mathbf{x}_i^t\|^2\right] + \frac{2}{Tn}\sum_{t=0}^{T-1}\sum_{i=1}^{n}\mathbb{E}\left[\|\nabla f(\bar{\mathbf{x}}^t)^2\|^2\right] \\
&= \frac{2L^2}{Tn}\sum_{t=0}^{T-1}\Omega_3^t + \frac{2}{Tn}\sum_{t=0}^{T-1}\sum_{i=1}^{n}\mathbb{E}\left[\|\nabla f(\bar{\mathbf{x}}^t)^2\|^2\right].
\end{aligned}
$$

## B.2 PŁ SETTING.

**Theorem 2** (Convergence of MoTEF). *Let Assumptions 2 to 4 hold. Then there exist absolute constants $c_\gamma, c_\lambda, c_\eta$, and some $\tau \leq 1$ such that if we set stepsizes $\gamma = c_\gamma \alpha \rho$, $\lambda = c_\lambda \alpha \rho^3 \tau$, $\eta = c_\eta L^{-1} \alpha \rho^3 \tau$, and choosing the initial batch size $B_{\text{init}} \geq \lceil \frac{LF^0}{\sigma^2} \rceil$, then after at most*

$$T = \widetilde{\mathcal{O}} \left( \frac{L\sigma^2}{\mu^2 n \varepsilon} + \frac{L\sigma}{\alpha \rho^{5/2} \mu^{3/2} \varepsilon^{1/2}} + \frac{L}{\mu \alpha \rho^3} \right) \tag{11}$$

*iterations of Algorithm 1 it holds $\mathbb{E}\left[ f(\mathbf{x}^T) - f^* \right] \leq \varepsilon$, and $\widetilde{\mathcal{O}}$ suppresses absolute constants and poly-logarithmic factors.*

*Proof.* The only change in the proof is the descent of the Lyapunov function. In PŁ case, the descent on $\Phi^t$ becomes

$$\Phi^{t+1} = F^{t+1} + \mathbf{b}^\top \mathbf{\Omega}^t$$

$$\leq F_t - \frac{\eta}{2} \mathbb{E}\left[ \left\| \nabla f(\bar{\mathbf{x}}^t) \right\|^2 \right] + \frac{\eta}{n^2} \hat{G}^t + \frac{\eta L^2}{n} \Omega_3^t - (\eta/2 - \eta^2 L/2)\Omega_5^t$$

$$+ \ \mathbf{b}^\top (\mathbf{A}\mathbf{\Omega}^t + \Omega_5^t \mathbf{b}_1 + \lambda^2 \sigma^2 \mathbf{b}_2)$$

$$\leq (1 - \eta\mu)F^t + (1 - \eta\mu)\mathbf{b}^\top \mathbf{\Omega}^t + (\mathbf{q}^\top + \mathbf{b}^\top \mathbf{A} - (1 - \eta\mu)\mathbf{b}^\top)\mathbf{\Omega}^t$$

$$- \ (\eta/2 - \eta^2 L/2 - \mathbf{b}^\top \mathbf{b}_1)\Omega_5^t + \mathbf{b}^\top \mathbf{b}_2 \lambda^2 \sigma^2$$

$$= (1 - \eta\mu)\Phi^t + (\mathbf{q}^\top + \mathbf{b}^\top \mathbf{A} - (1 - \eta\mu)\mathbf{b}^\top)\mathbf{\Omega}^t - (\eta/2 - \eta^2 L/2 - \mathbf{b}^\top \mathbf{b}_1)\Omega_5^t + \mathbf{b}^\top \mathbf{b}_2 \lambda^2 \sigma^2,$$

where in the second inequality we use PŁ condition. Similar to the proof of Theorem 1, we need to satisfy

$$\begin{bmatrix} (1 - \mu\eta)\mathbf{I} - \mathbf{A}^\top \\ -\mathbf{b}_1^\top \end{bmatrix} \mathbf{b} \geq \begin{bmatrix} \mathbf{q} \\ \frac{\eta^2 L}{2} - \frac{\eta}{2} \end{bmatrix}$$

for some coefficients $\mathbf{b}$. We set the stepsizes such that

$$\lambda := c_\lambda \alpha \rho^3 \tau, \quad \gamma := c_\gamma \alpha \rho, \quad \eta := \frac{c_\eta \alpha \rho^3 \tau}{L},$$

and

$$\mathbf{b} := \left( \frac{b_1}{n^2 L}, \frac{b_2 \tau}{nL}, \frac{b_3 L}{\rho^3 n \tau}, \frac{b_4 \tau}{\rho n L}, \frac{b_5 L}{\rho^3 n \tau}, \frac{b_6 \tau}{\rho n L} \right)^\top$$

with the choice

$$c_\lambda = \frac{1}{200000}, c_\gamma = \frac{1}{200000}, c_\eta = \frac{1}{100000000},$$

and

$$b_1 = \frac{1}{250}, b_2 = \frac{13}{200000}, b_3 = \frac{1}{20}, b_4 = \frac{1}{400000}, b_5 = 2, b_6 = \frac{1}{200000},$$

gives the following descent on $\Phi^t$ (note that both $\gamma$ and $\lambda$ are smaller than 1 with this choice of constants)

$$\Phi^{t+1} \ \leq \ \left( 1 - \frac{c_\eta \alpha \rho^3 \tau \mu}{L} \right) \Phi^t + \frac{c_\lambda^2 b_1 \alpha^2 \rho^6}{nL} \cdot \tau^2 \sigma^2$$

$$+ \left( \frac{6 c_\lambda^2 b_4 \alpha \rho^5}{L} + \frac{2 c_\lambda^2 b_2 \alpha^2 \rho^6}{L} + \frac{6 b_6 c_\lambda^2 \alpha \rho^4}{c_\gamma L} \right) \tau^3 \sigma^2. \tag{37}$$

Let us define constants

$$B \ := \ \frac{c_\eta \alpha \rho^3 \mu}{2L},$$

$$C \ := \ \frac{c_\lambda^2 c_1 \alpha^2 \rho^6}{nL},$$

$$D \ := \ \left( \frac{6 c_\lambda^2 c_4 \alpha \rho^5}{L} + \frac{2 c_\lambda^2 c_2 \alpha^2 \rho^6}{L} + \frac{6 c_6 c_\lambda^2 \alpha \rho^4}{c_\gamma L} \right),$$

$$E \ := \ 1.$$

Unrolling (37) for $T$ iterations we get

$$\Phi^T \le (1 - B\tau)^T \Phi^0 + \frac{C}{B\tau}\tau^2\sigma^2 + \frac{D}{B\tau}\tau^3\sigma^2 = (1 - B\tau)^T \Phi^0 + \frac{C}{B}\sigma^2 + \frac{D}{B\tau}\tau^3\sigma^2$$

where we use the fact that

$$\sum_{l=0}^{m-1}(1 - B\tau)^l = \frac{1 - (1 - B\tau)^m}{1 - (1 - B\tau)} \le \frac{1}{B\tau}.$$

Choosing $\tau = \min\left\{ \frac{1}{E}, \frac{1}{BT}\log\left(\frac{\Phi^0 B^2 T}{C\sigma^2}\right), \frac{1}{BT}\log\left(\frac{\Phi^0 B^3 T^2}{D\sigma^2}\right) \right\}$ leads to the following rate

$$\Phi^T \le \widetilde{\mathcal{O}}\left( \exp\left(-\frac{B}{E}T\right)\Phi^0 + \frac{C\sigma^2}{B^2 T} + \frac{D\sigma^2}{B^3 T^2} \right).$$

We refer to (Mishchenko et al., 2020) for a more detailed derivation (proof of Corollary 1, page 20). To achieve $F^T \le \varepsilon$, we need to perform

$$
\begin{aligned}
T &= \widetilde{\mathcal{O}}\left( \frac{E}{B} + \frac{C\sigma^2}{B^2\varepsilon} + \frac{\sqrt{D}\sigma}{B^{3/2}\varepsilon^{1/2}} \right) \\
&= \widetilde{\mathcal{O}}\left( \frac{L}{\mu\alpha\rho^3} + \frac{L\sigma^2}{\mu^2 n\varepsilon} + \frac{L\sigma}{\alpha^{1/2}\rho^2\mu^{3/2}\varepsilon^{1/2}} + \frac{L\sigma}{\alpha\rho^{5/2}\mu^{3/2}\varepsilon^{1/2}} + \frac{L\sigma}{\alpha\rho^2\mu^{3/2}\varepsilon^{1/2}} \right).
\end{aligned}
$$

iterations. Note that the fourth term always dominates the third and fifth terms. Therefore, we remove them from the rate and derive the following rate

$$T = \widetilde{\mathcal{O}}\left( \frac{L}{\mu\alpha\rho^3} + \frac{L\sigma^2}{\mu^2 n\varepsilon} + \frac{L\sigma}{\alpha\rho^{5/2}\mu^{3/2}\varepsilon^{1/2}} \right).$$

$\square$

## C  MISSING PROOFS FOR MoTEF-VR

In this section, we provide the proof of convergence of Algorithm 2. Note that in this case Lemma 7 remains unchanged.

**Lemma 17.** *Let Assumptions 3 and 5 hold. Then we have the following descent on $\hat{G}^t$*

$$\hat{G}^{t+1} \le (1 - \lambda)\hat{G}^t + 2\lambda^2\sigma^2 n + \ell^2\mathbb{E}\left[\|\mathbf{X}^{t+1} - \mathbf{X}^t\|_F^2\right]. \tag{38}$$

*Proof.* We have

$$
\begin{aligned}
\hat{G}^{t+1} &= \mathbb{E}\left[\left\|\mathbf{M}^{t+1}\mathbf{1} - \nabla F(\mathbf{X}^{t+1})\mathbf{1}\right\|^2\right] \\
&= \mathbb{E}\left[\left\|[\widetilde{\nabla}F(\mathbf{X}^{t+1}, \Xi^{t+1}) + (1 - \lambda)(\mathbf{M}^t - \widetilde{\nabla}F(\mathbf{X}^t, \Xi^{t+1}) - \nabla F(\mathbf{X}^{t+1})]\mathbf{1}\right\|^2\right] \\
&= \mathbb{E}\left[\left\|\left(\lambda(\widetilde{\nabla}F(\mathbf{X}^{t+1}, \Xi^{t+1}) - \nabla F(\mathbf{X}^{t+1}))\right.\right.\right. \\
&\qquad\qquad + (1 - \lambda)(\widetilde{\nabla}F(\mathbf{X}^{t+1}, \Xi^{t+1}) - \nabla F(\mathbf{X}^{t+1}) + \nabla F(\mathbf{X}^t) - \widetilde{\nabla}F(\mathbf{X}^t, \Xi^{t+1})) \\
&\qquad\qquad \left.\left.\left. + (1 - \lambda)(\mathbf{M}^t - \nabla F(\mathbf{X}^t))\right)\mathbf{1}\right\|^2\right] \\
&\le (1 - \lambda)^2\,\mathbb{E}\left\|(\mathbf{M}^t - \nabla F(\mathbf{X}^t))\mathbf{1}\right\|^2 \\
&\qquad + 2\lambda^2\,\mathbb{E}\left\|(\widetilde{\nabla}F(\mathbf{X}^{t+1}, \Xi^{t+1}) - \nabla F(\mathbf{X}^{t+1}))\mathbf{1}\right\|^2 \\
&\qquad + 2(1 - \lambda)^2\,\mathbb{E}\left\|(\widetilde{\nabla}F(\mathbf{X}^{t+1}, \Xi^{t+1} - \nabla F(\mathbf{X}^{t+1}) + \nabla F(\mathbf{X}^t) - \widetilde{\nabla}F(\mathbf{X}^t, \Xi^{t+1}))\mathbf{1}\right\|^2 \\
&\le (1 - \lambda)\hat{G}^t + 2\lambda^2\sigma^2 n \\
&\qquad + 2\,\mathbb{E}\left\|(\widetilde{\nabla}F(\mathbf{X}^{t+1}, \Xi^{t+1}) - \nabla F(\mathbf{X}^{t+1}) + \nabla F(\mathbf{X}^t) - \widetilde{\nabla}F(\mathbf{X}^t, \Xi^{t+1}))\mathbf{1}\right\|^2. \tag{39}
\end{aligned}
$$

For the last term above we continue as follows

$$\mathbb{E}\left[\|(\widetilde{\nabla}F(\mathbf{X}^{t+1}, \Xi^{t+1}) - \nabla F(\mathbf{X}^{t+1}) + \nabla F(\mathbf{X}^t) - \widetilde{\nabla}F(\mathbf{X}^t, \Xi^{t+1}))\mathbf{1}\|^2\right]$$

$$= \mathbb{E}\left[\left\|\sum_{i=1}^n \nabla f_i(\mathbf{x}_i^{t+1}, \xi_i^{t+1}) - \nabla f_i(\mathbf{x}_i^{t+1}) + \nabla f_i(\mathbf{x}_i^t) - \nabla f_i(\mathbf{x}_i^t, \xi_i^{t+1})\right\|^2\right]$$

$$= \sum_{i=1}^n \mathbb{E}\left[\left\|\nabla f_i(\mathbf{x}_i^{t+1}, \xi_i^{t+1}) - \nabla f_i(\mathbf{x}_i^{t+1}) + \nabla f_i(\mathbf{x}_i^t) - \nabla f_i(\mathbf{x}_i^t, \xi_i^{t+1})\right\|^2\right]$$

$$\leq \sum_{i=1}^n \mathbb{E}\left[\left\|\nabla f_i(\mathbf{x}_i^{t+1}, \xi_i^{t+1}) - \nabla f_i(\mathbf{x}_i^t, \xi_i^{t+1})\right\|^2\right]$$

$$\leq \ell^2 \mathbb{E}\left[\|\mathbf{X}^{t+1} - \mathbf{X}^t\|_{\mathrm{F}}^2\right]. \tag{40}$$

Therefore, from (39) we get

$$\hat{G}^{t+1} \leq (1-\lambda)\hat{G}^t + 2\lambda^2\sigma^2 n + \ell^2\mathbb{E}\left[\|\mathbf{X}^{t+1} - \mathbf{X}^t\|_{\mathrm{F}}^2\right]. \tag{41}$$

$\square$

**Lemma 18.** *Assume Assumptions 3 and 5 hold. Then we have the following descent on $\hat{G}^t$*

$$\widetilde{G}^{t+1} \leq (1-\lambda)\widetilde{G}^t + 2\lambda^2\sigma^2 n + \ell^2\mathbb{E}\left[\|\mathbf{X}^{t+1} - \mathbf{X}^t\|_{\mathrm{F}}^2\right]. \tag{42}$$

*Proof.* The proof is similar to the one of Lemma 17. $\square$

Note that Lemmas 10 to 14 and 16 do not change in this setting, thus, we do not repeat them.

**Lemma 19.** *Assume Assumptions 3 and 5 hold. Then we have the following control of momentum at iterations $t$ and $t+1$*

$$\mathbb{E}\left[\|\mathbf{M}^{t+1} - \mathbf{M}^t\|_{\mathrm{F}}^2\right] \leq \lambda^2\widetilde{G}^t + 2\lambda^2 n\sigma^2 + 2\ell^2\mathbb{E}\left[\|\mathbf{X}^{t+1} - \mathbf{X}^t\|_{\mathrm{F}}^2\right]. \tag{43}$$

*Proof.* Using the update of $\mathbf{M}^t$ we have

$$\mathbb{E}\left[\|\mathbf{M}^{t+1} - \mathbf{M}^t\|_{\mathrm{F}}^2\right] = \mathbb{E}\left[\|\widetilde{\nabla}F(\mathbf{X}^{t+1}, \Xi^{t+1}) + (1-\lambda)(\mathbf{M}^t - \widetilde{\nabla}F(\mathbf{X}^t, \Xi^{t+1})) - \mathbf{M}^t\|_{\mathrm{F}}^2\right]$$

$$= \mathbb{E}\left[\|\widetilde{\nabla}F(\mathbf{X}^{t+1}, \Xi^{t+1}) - \lambda\mathbf{M}^t - (1-\lambda)\widetilde{\nabla}F(\mathbf{X}^t, \Xi^{t+1})\|_{\mathrm{F}}^2\right]$$

$$= \mathbb{E}\left[\left\|\lambda(\nabla F(\mathbf{X}^t) - \mathbf{M}^t) + \lambda(\widetilde{\nabla}F(\mathbf{X}^t, \Xi^{t+1}) - \nabla F(\mathbf{X}^t))\right.\right.$$

$$\left.\left. + (\widetilde{\nabla}F(\mathbf{X}^{t+1}, \Xi^{t+1}) - \widetilde{\nabla}F(\mathbf{X}^t, \Xi^{t+1}))\right\|_{\mathrm{F}}^2\right]$$

$$= \lambda^2\widetilde{G}^t + \mathbb{E}\left[\left\|\lambda(\widetilde{\nabla}F(\mathbf{X}^t, \Xi^{t+1}) - \nabla F(\mathbf{X}^t))\right.\right.$$

$$\left.\left. + (\widetilde{\nabla}F(\mathbf{X}^{t+1}, \Xi^{t+1}) - \widetilde{\nabla}F(\mathbf{X}^t, \Xi^{t+1}))\right\|_{\mathrm{F}}^2\right]$$

$$\leq \lambda^2\widetilde{G}^t + 2\lambda^2 n\sigma^2 + 2\ell^2\mathbb{E}\left[\|\mathbf{X}^{t+1} - \mathbf{X}^t\|_{\mathrm{F}}^2\right].$$

$\square$

Now we introduce the following Lyapunov function of the form

$$\Psi^t := F^t + \frac{d_1}{\alpha\rho^3 n\tau\ell}\hat{G}^t + \frac{d_2}{n\ell}\widetilde{G}^t + \frac{d_3\ell}{\rho^3 n\tau}\Omega_1^t + \frac{d_4}{\rho n\ell}\Omega_2^t + \frac{d_5\ell}{\rho^3 n\tau}\Omega_3^t + \frac{d_6}{\rho n\ell}\Omega_4^t, \tag{44}$$

where $\{d_k\}_{k=1}^6$ are absolute constants defined in (49). Again, we present the descent lemma on the Lyapunov function $\Psi^t$.

**Lemma 20** (Descent of the Lyapunov function). *Let Assumptions 3 and 5 hold. Then there exists absolute constants $c_\gamma, c_\lambda, c_\eta$ and $\tau < 1$ such that if we set stepsizes $\gamma = c_\gamma \alpha \rho$, $\lambda = c_\lambda n^{-1} \alpha^2 \rho^6 \tau^2$, $\eta = c_\eta \ell^{-1} \alpha \rho^3 \tau$ then the Lyapunov function $\Psi^t$ decreases as*

$$
\begin{aligned}
\Psi^{t+1} \quad \leq \quad & \Psi^t - \frac{c_\eta \alpha \rho^3}{2\ell} \tau \mathbb{E}\left[\|\nabla f(\bar{\mathbf{x}}^t)\|^2\right] + \frac{2c_1 c_\lambda^2}{n^2 \ell} \alpha^3 \rho^9 \tau^3 \sigma^2 \\
& + \left(\frac{2d_2 c_\lambda^2 \alpha^4 \rho^{12}}{n^2 \ell} + \frac{12 d_4 c_\lambda^2 \alpha^3 \rho^{11}}{n^3 \ell} + \frac{6 d_6 c_\lambda^2 \alpha^3 \rho^{10}}{n^3 \ell}\right) \tau^4 \sigma^2.
\end{aligned}
\tag{45}
$$

**Remark 21.** *Compared to Lemma 1, in Lemma 20, the leading stochastic term has a cubic dependence on $\tau$, whereas in Lemma 1 the dependence is quadratic. The improved dependence on $\tau$ is the key ingredient to the speed-up for variance reduction type methods.*

*Proof.* From Lemmas 14 and 19 we get

$$
\begin{aligned}
\mathbb{E}\left[\|\mathbf{M}^{t+1} - \mathbf{M}^t\|_{\mathrm{F}}^2\right] &\leq \lambda^2 \widetilde{G}^t + 2\lambda^2 n \sigma^2 + 2\ell^2 (3\gamma^2 C \Omega_1^t + 3\gamma^2 C \Omega_3^t + 3\eta^2 \Omega_4^t + 3\eta^2 n \Omega_5^t) \\
&= 2\lambda^2 n \sigma^2 + \lambda^2 \widetilde{G}^t + 6C\gamma^2 \ell^2 \Omega_1^t + 6C\gamma^2 \ell^2 \Omega_3^t + 6\eta^2 \ell^2 \Omega_4^t + 6n\eta^2 \ell^2 \Omega_5^t.
\end{aligned}
\tag{46}
$$

From the above inequality (46) and Lemma 16 we get

$$
\begin{aligned}
\mathbb{E}\left[\|\mathbf{V}^{t+1} - \mathbf{V}^t\|_{\mathrm{F}}^2\right] &\leq 3\gamma^2 C \Omega_2^t + 3\gamma^2 C \Omega_4^t \\
&\quad + 3\left(2\lambda^2 n \sigma^2 + \lambda^2 \widetilde{G}^t + 6C\gamma^2 \ell^2 \Omega_1^t + 6C\gamma^2 \ell^2 \Omega_3^t + 6\eta^2 \ell^2 \Omega_4^t + 6n\eta^2 \ell^2 \Omega_5^t\right) \\
&= 6\lambda^2 n \sigma^2 + 3\lambda^2 \widetilde{G}^t + 18C\gamma^2 \ell^2 \Omega_1^t + 3C\gamma^2 \Omega_2^t + 18C\gamma^2 \ell^2 \Omega_3^t \\
&\quad + (3C\gamma^2 + 18\eta^2 \ell^2)\Omega_4^t + 18n\eta^2 \ell^2 \Omega_5^t.
\end{aligned}
\tag{47}
$$

From Lemmas 14 and 17 we get the following descent on $\hat{G}^t$

$$
\begin{aligned}
\hat{G}^{t+1} &\leq (1-\lambda)\hat{G}^t + 2\lambda^2 \sigma^2 n + \ell^2(3\gamma^2 C \Omega_1^t + 3\gamma^2 C \Omega_3^t + 3\eta^2 \Omega_4^t + 3\eta^2 n \Omega_5^t) \\
&= 2\lambda^2 \sigma^2 n + (1-\lambda)\hat{G}^t + 3C\gamma^2 \ell^2 \Omega_1^t + 3C\gamma^2 \ell^2 \Omega_3^t + 3\eta^2 \ell^2 \Omega_4^t + 3n\eta^2 \ell^2 \Omega_5^t.
\end{aligned}
$$

Similarly, from Lemmas 14 and 18 we get the following descent on $\widetilde{G}^t$

$$
\widetilde{G}^{t+1} \leq 2\lambda^2 \sigma^2 n + (1-\lambda)\widetilde{G}^t + 3C\gamma^2 \ell^2 \Omega_1^t + 3C\gamma^2 \ell^2 \Omega_3^t + 3\eta^2 \ell^2 \Omega_4^t + 3n\eta^2 \ell^2 \Omega_5^t.
$$

From Lemmas 10 and 14 we get the following descent on $\Omega_1^t$

$$
\begin{aligned}
\Omega_1^{t+1} &\leq (1-\alpha/2)\mathbb{E}\left[\|\mathbf{H}^t - \mathbf{X}^t\|_{\mathrm{F}}^2\right] + \frac{2}{\alpha}(3\gamma^2 C \Omega_1^t + 3\gamma^2 C \Omega_3^t + 3\eta^2 \Omega_4^t + 3\eta^2 n \Omega_5^t) \\
&= (1-\alpha/2 + 6C\gamma^2/\alpha)\Omega_1^t + \frac{6C\gamma^2}{\alpha}\Omega_3^t + \frac{6\eta^2}{\alpha}\Omega_4^t + \frac{6n\eta^2}{\alpha}\Omega_5^t.
\end{aligned}
$$

From Lemma 11 and (47) we get the following descent on $\Omega_2^t$

$$
\begin{aligned}
\Omega_2^{t+1} &\leq (1-\alpha/2)\Omega_2^t + \frac{2}{\alpha}\bigg(6\lambda^2 n \sigma^2 + 3\lambda^2 \widetilde{G}^t + 18C\gamma^2 \ell^2 \Omega_1^t + 3C\gamma^2 \Omega_2^t + 18C\gamma^2 \ell^2 \Omega_3^t \\
&\quad + (3C\gamma^2 + 18C\eta^2 \ell^2)\Omega_4^t + 18n\eta^2 \ell^2 \Omega_5^t.\bigg) \\
&= \frac{12n\lambda^2}{\alpha}\sigma^2 + \frac{6\lambda^2}{\alpha}\widetilde{G}^t + \frac{36C\gamma^2 \ell^2}{\alpha}\Omega_1^t + (1-\alpha/2 + 6C\gamma^2/\alpha)\Omega_2^t + \frac{36C\gamma^2 \ell^2}{\alpha}\Omega_3^t \\
&\quad + \frac{2}{\alpha}(3C\gamma^2 + 18\eta^2 \ell^2)\Omega_4^t + \frac{36n\eta^2 \ell^2}{\alpha}\Omega_5^t.
\end{aligned}
$$

The descent on $\Omega_3^t$ (29) from the proof of MoTEF remains unchanged

$$\Omega_3^{t+1} \leq (1 - \frac{\gamma\rho}{2})\Omega_3^t + \frac{6\gamma C}{\rho}\Omega_1^t + \frac{6\eta^2}{\gamma\rho}\Omega_4^t.$$

From Lemma 13 and (46) we get the following descent on $\Omega_4^t$

$$
\begin{aligned}
\Omega_4^{t+1} &\leq (1 - \gamma\rho/2)\Omega_4^t + 2\gamma^2 C(1 + 2/\gamma\rho)\Omega_2^t \\
&\quad + 2(1 + 2/\gamma\rho)(2\lambda^2 n\sigma^2 + \lambda^2 \widetilde{G}^t + 6C\gamma^2\ell^2\Omega_1^t + 6C\gamma^2\ell^2\Omega_3^t + 6\eta^2\ell^2\Omega_4^t + 6n\eta^2\Omega_5^t) \\
&\leq \frac{6n\lambda^2}{\gamma\rho}\sigma^2 + \frac{3\lambda^2}{\gamma\rho}\widetilde{G}^t + \frac{18C\gamma\ell^2}{\rho}\Omega_1^t + \frac{6C\gamma}{\rho}\Omega_2^t + \frac{18C\gamma\ell^2}{\rho}\Omega_3^t + (1 - \gamma\rho/2 + 18\eta^2\ell^2/\gamma\rho)\Omega_4^t \\
&\quad + \frac{18n\eta^2\ell^2}{\gamma\rho}\Omega_5^t.
\end{aligned}
$$

We remind that $\boldsymbol{\Omega} = (\hat{G}^t, \widetilde{G}^t, \Omega_1^t \Omega_2^t, \Omega_3^t, \Omega_4^t)^\top$. Now we can gather all inequalities together

$$
\boldsymbol{\Omega}^{t+1} \leq \underbrace{\begin{pmatrix}
1-\lambda & 0 & 3C\gamma^2\ell^2 & 0 & 3C\gamma^2\ell^2 & 3\eta^2\ell^2 \\
0 & 1-\lambda & 3C\gamma^2\ell^2 & 0 & 3C\gamma^2\ell^2 & 3\eta^2\ell^2 \\
0 & 0 & 1-\frac{\alpha}{2}+\frac{6C\gamma^2}{\alpha} & 0 & \frac{6C\gamma^2}{\alpha} & \frac{6\eta^2}{\alpha} \\
0 & \frac{6\lambda^2}{\alpha} & \frac{36C\gamma^2\ell^2}{\alpha} & 1-\frac{\alpha}{2}+\frac{6C\gamma^2}{\alpha} & \frac{36C\gamma^2\ell^2}{\alpha} & \frac{6C\gamma^2}{\alpha}+\frac{36\eta^2\ell^2}{\alpha} \\
0 & 0 & \frac{6\gamma C}{\rho} & 0 & 1-\frac{\gamma\rho}{2} & \frac{6\eta^2}{\gamma\rho} \\
0 & \frac{3\lambda^2}{\gamma\rho} & \frac{18C\gamma\ell^2}{\rho} & \frac{6C\gamma}{\rho} & \frac{18C\gamma\ell^2}{\rho} & 1-\frac{\gamma\rho}{2}+\frac{18\eta^2\ell^2}{\gamma\rho}
\end{pmatrix}}_{:=\mathbf{A}} \boldsymbol{\Omega}^t
$$

$$
+ \underbrace{\begin{pmatrix}
3n\eta^2\ell^2 \\
3n\eta^2\ell^2 \\
\frac{6n\eta^2}{\alpha} \\
\frac{36n\eta^2\ell^2}{\alpha} \\
0 \\
\frac{18n\eta^2\ell^2}{\gamma\rho}
\end{pmatrix}}_{:=\mathbf{b}_1} \Omega_5^t + \underbrace{\begin{pmatrix}
2n \\
2n \\
0 \\
\frac{12n}{\alpha} \\
0 \\
\frac{6n}{\gamma\rho}
\end{pmatrix}}_{:=\mathbf{b}_2} \lambda^2\sigma^2. \tag{48}
$$

Now we consider the following choice of stepsizes

$$\lambda := \frac{c_\lambda \alpha^2 \rho^6 \tau^2}{n}, \quad \gamma := c_\gamma \alpha\rho, \quad \eta := \frac{c_\eta \alpha\rho^3\tau}{\ell},$$

and constants

$$\mathbf{d} := \left( \frac{d_1}{\alpha\rho^3 n\tau\ell}, \frac{d_2}{n\ell}, \frac{d_3\ell}{\rho^3 n\tau}, \frac{d_4}{\rho n\ell}, \frac{d_5\ell}{\rho^3 n\tau}, \frac{d_6}{\rho n\ell} \right)^\top,$$

where

$$c_\lambda = \frac{1}{200}, c_\gamma = \frac{1}{200}, c_\eta = \frac{1}{100000},$$
$$d_1 = 0.0020, d_2 = 0.000065, d_3 = 0.005, d_4 = 0.0000025, d_5 = 0.01, d_6 = 0.000005 \tag{49}$$

Note that choosing $\tau \leq 1$ makes the system of inequalities (48) hold. Using this choice, we get the following descent on $\Psi^t = F^t + \mathbf{d}^\top \boldsymbol{\Omega}^t$

$$\Psi^{t+1} \leq \Psi^t - \frac{c_\eta \alpha \rho^3 \tau}{2\ell} \mathbb{E}\left[\|\nabla f(\bar{\mathbf{x}}^t)\|^2\right] + \frac{d_1}{\alpha \rho^3 n \tau \ell} \cdot 2n c_\lambda^2 \alpha^4 \rho^{12} n^{-2} \tau^4 \sigma^2$$

$$+ \frac{d_2}{n\ell} \cdot 2n c_\lambda^2 \alpha^4 \rho^{12} n^{-2} \tau^4 \sigma^2$$

$$+ \frac{d_4}{\rho n\ell} \cdot \frac{12n}{\alpha} c_\lambda^2 \alpha^4 \rho^{12} n^{-2} \tau^4 \sigma^2$$

$$+ \frac{d_6}{\rho n\ell} \cdot \frac{6n}{c_\gamma \alpha \rho} c_\lambda^2 \alpha^4 \rho^{12} \tau^4 n^{-2} \sigma^2$$

$$= \Psi^t - \frac{c_\eta \alpha \rho^3}{2\ell} \tau \mathbb{E}\left[\|\nabla f(\bar{\mathbf{x}}^t)\|^2\right] + \frac{2 d_1 c_\lambda^2}{n^2 \ell} \alpha^3 \rho^9 \tau^3 \sigma^2$$

$$+ \left(\frac{2 d_2 c_\lambda^2 \alpha^4 \rho^{12}}{n^2 \ell} + \frac{12 d_4 c_\lambda^2 \alpha^3 \rho^{11}}{n^2 \ell} + \frac{6 d_6 c_\lambda^2 \alpha^3 \rho^{10}}{n^2 \ell}\right) \tau^4 \sigma^2. \tag{50}$$

By this, we proved Lemma 20. $\qquad\square$

**Theorem 3** (Convergence of MoTEF-VR). *Let Assumptions 3 and 5 hold. Then there exists absolute constants $c_\gamma, c_\lambda, c_\eta$ and some $\tau < 1$ such that if we stepsizes $\gamma = c_\gamma \alpha \rho, \lambda = c_\lambda n^{-1} \alpha^2 \rho^6 \tau^2, \eta = c_\eta \ell^{-1} \alpha \rho^3 \tau$, and initial batch size $B_{\mathrm{init}} \geq \lceil \frac{\sigma^2}{LF^0 \alpha \rho^3} \rceil$, then after at most*

$$T = \mathcal{O}\left(\frac{\sigma}{n\varepsilon^3} + \frac{\sigma^{2/3}}{n^{2/3} \alpha^{1/3} \rho^{2/3} \varepsilon^{8/3}} + \frac{1}{\alpha \rho^3 \varepsilon^2}\right) \ell F^0 \tag{13}$$

*iterations of Algorithm 2 it holds $\mathbb{E}\left[\|\nabla f(\mathbf{x}_{\mathrm{out}})\|^2\right] \leq \varepsilon^2$, where $\mathbf{x}_{\mathrm{out}}$ is chosen uniformly at random from $\{\bar{\mathbf{x}}_0, \cdots, \bar{\mathbf{x}}_{T-1}\}$, and $\mathcal{O}$ suppresses absolute constants and poly-logarithmic factors.*

*Proof.* We apply Lemma 20 and consider the following:

$$B := \frac{c_\eta \alpha \rho^3}{2\ell},$$

$$C := \frac{2 d_1 c_\lambda^2}{n^2 \ell} \alpha^3 \rho^9,$$

$$D := \left(\frac{2 d_2 c_\lambda^2 \alpha^4 \rho^{12}}{n^2 \ell} + \frac{12 d_4 c_\lambda^2 \alpha^3 \rho^{11}}{n^2 \ell} + \frac{6 d_6 c_\lambda^2 \alpha^3 \rho^{10}}{n^2 \ell}\right),$$

$$E := 1.$$

Unrolling (50) for $T$ iterations we get

$$\frac{1}{T} \sum_{t=0}^{T-1} \mathbb{E}\left[\|\nabla f(\bar{\mathbf{x}}^t)\|^2\right] \leq \frac{\Phi^0}{\tau B T} + \frac{C}{B} \tau^2 \sigma^2 + \frac{D}{B} \tau^3 \sigma^2.$$

Choosing $\tau = \min\left\{\frac{1}{E}, \left(\frac{\Psi^0}{C\sigma^2 T}\right)^{1/3}, \left(\frac{\Psi^0}{D\sigma^2 T}\right)^{1/4}\right\}$ gives the following rate

$$\frac{1}{T} \sum_{t=0}^{T-1} \mathbb{E}\left[\|\nabla f(\bar{\mathbf{x}}^t)\|^2\right] \leq \frac{E\Psi^0}{BT} + \left(\frac{\sqrt{C}\Psi^0 \sigma}{B^{3/2} T}\right)^{2/3} + \left(\frac{D^{1/3}\Psi^0 \sigma^{2/3}}{B^{4/3} T}\right)^{3/4}$$

$$= \mathcal{O}\left(\frac{\ell\Psi^0}{\alpha \rho^3 T} + \left(\frac{\ell\Psi^0 \sigma}{nT}\right)^{2/3}\right.$$

$$\left. + \left(\frac{(n^{-2/3} + \alpha^{-1/3}\rho^{-1/3} n^{-2/3} + \alpha^{-1/3}\rho^{-2/3} n^{-2/3})\ell\Psi^0 \sigma^{2/3}}{T}\right)^{3/4}\right),$$

that translates into the rate in terms of $\varepsilon$ to

$$\frac{1}{T} \sum_{t=0}^{T-1} \mathbb{E}\left[\|\nabla f(\bar{\mathbf{x}}^t)\|^2\right] \leq \varepsilon^2 \Rightarrow \mathcal{O}\left(\frac{\ell\Psi^0}{\alpha \rho^3 \varepsilon^2} + \frac{\ell\Psi^0 \sigma}{n\varepsilon^3} + \frac{\ell\Psi^0 \sigma^{2/3}}{n^{2/3} \varepsilon^{8/3}} + \frac{\ell\Psi^0 \sigma^{2/3}}{\alpha^{1/3} \rho^{1/3} n^{2/3} \varepsilon^{8/3}}\right.$$

$$\left. + \frac{\ell\Psi^0 \sigma^{2/3}}{\alpha^{1/3} \rho^{2/3} n^{2/3} \varepsilon^{8/3}}\right).$$

Note that the last term always dominates the third and fourth terms in the rate. Therefore, the final convergence rate has the following form

$$\frac{1}{T}\sum_{t=0}^{T-1}\mathbb{E}\left[\|\nabla f(\bar{\mathbf{x}}^t)\|^2\right] \leq \varepsilon^2 \quad \Rightarrow \quad \mathcal{O}\left(\frac{\ell\Psi^0}{\alpha\rho^3\varepsilon^2} + \frac{\ell\Psi^0\sigma}{n\varepsilon^3} + \frac{\ell\Psi^0\sigma^{2/3}}{\alpha^{1/3}\rho^{2/3}n^{2/3}\varepsilon^{8/3}}\right).$$

Note that with the choice $\mathbf{V}^0 = \mathbf{G}^0 = \mathbf{M}^0 = \widetilde{\nabla}F(\mathbf{X}^0), \mathbf{H}^0 = \mathbf{X}^0 = \mathbf{x}^0\mathbf{1}^\top$, we get

$$\hat{G}^0 \leq \sigma^2 n, \quad \widetilde{G}^0 \leq \sigma^2 n, \quad \Omega_1^0 = \Omega_2^0 = \Omega_3^0 = \Omega_4^0 = 0.$$

$$\Psi^0 \leq F^0 + \frac{d_1}{\alpha\rho^3 n\tau\ell}\sigma^2 n + \frac{d_2}{n\ell}\sigma^2 n. \tag{51}$$

If we choose the initial batch size $B_{\text{init}} \geq \lceil\frac{\sigma^2}{LF^0\alpha\rho^3}\rceil$, we get

$$\Psi^0 \leq F^0 + \frac{1}{\alpha\rho^3\ell}\frac{\sigma^2}{B_{\text{init}}} + \frac{1}{\ell}\frac{\sigma^2}{B_{\text{init}}} \leq 3F^0. \tag{52}$$

$\square$

# D EXPERIMENT DETAILS

## D.1 EFFECT OF CHANGING HETEROGENEITY

We perform a grid search for the parameters $\gamma$ from $\{0.1, 0.01, 0.001\}$, $\eta$ from the log space from $10^{-4}$ to $10^{-1}$ and the log space from $5 \times 10^{-4}$ to $5 \times 10^{-1}$. For MoTEF we search the momentum parameter $\lambda$ from the same log space as $\eta$ as well.

## D.2 EFFECT OF COMMUNICATION TOPOLOGY (SYNTHETIC PROBLEM)

To study networks with different spectral gaps, we set $n = 400$ and construct random regular graphs with different degrees $r$. We sample the random graphs with degree $r \in \{3, 3, 3, 4, 4, 4, 4, 5, 5, 6, 6, 7, 10, 13, 16\}$, the resulting inverses of the spectral gaps are around $1/\rho \in \{21.41, 18.40, 18.59, 8.24, 8.55, 8.65, 7.92, 5.57, 5.36, 4.03, 4.34, 3.76, 2.56, 2.17, 1.99\}$.

## D.3 ROBUSTNESS TO COMMUNICATION TOPOLOGY.

Next, we study the effect of the network topology on the convergence of MoTEF. We set $n = 40, \lambda = 0.05$, choose batch size 100, and run experiments for ring, star, grid, Erdös-Rènyi ($p = 0.2$ and $p = 0.5$) topologies. For all topologies, we use $\eta = 0.05, \gamma = 0.5, \lambda = 0.01$ for a9a dataset and $\eta = 0.05, \gamma = 0.5, \lambda = 0.01$ for w8a. Note that the spectral gaps of these networks $0.012, 0.049, 0.063, 0.467, 0.755$ correspondingly.

## D.4 HYPERPARAMETERS FOR SECTION 4.2

For MoTEF we tune stepsize as follows $\eta \in \{0.001, 0.01, 0.05\}, \gamma \in \{0.1, 0.2, 0.5, 0.9\}, \lambda \in \{0.005, 0.01, 0.05, 0.1\}$. For BEER we tune the stepsizes in the range $\eta \in \{0.001, 0.01, 0.05\}, \gamma \in \{0.1, 0.2, 0.5, 0.9\}$. For Choco-SGD we tune the stepsizes in the range $\eta \in \{0.01, 0.05\}, \gamma \in \{0.1, 0.5, 0.9\}$. Finally, for DSGD and D2 we choose the stepsize $\eta = 0.01$.

## D.5 COMPARISON ON TRAINING CNN MODEL

We additionally provide an experiment where we compare MoTEF against BEER and Choco-SGD on ring and ER $p = 0.6$ topologies using CNN model on MNIST dataset. We use CNN model with two convolution layers each followed by batch normalization, ReLU, and max pooling. The classification layer is fully connected. We tune the stepsize for each algorithm from $\eta \in \{0.0001, 0.001, 0.01, 0.1\}$ and gossip stepsize $\gamma \in \{0.1, 0.9\}$. We demonstrate the performance of algorithms in Figure 6. We observe that in both cases MoTEF achieves faster convergence w.r.t. both test accuracy and train loss than competitors supporting our theoretical findings.

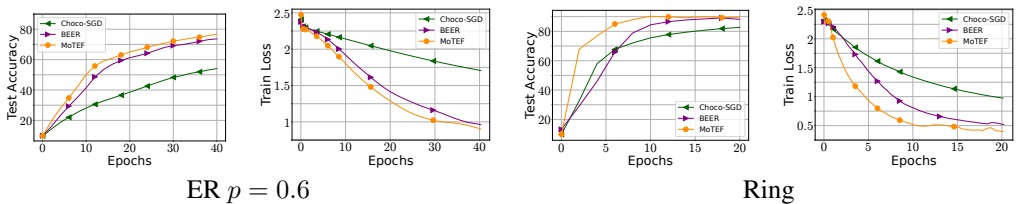

Figure 6: Performance of MoTEF, BEER, and Choco-SGD on ring and ER $p = 0.6$ topologies in training CNN model on MNIST dataset.

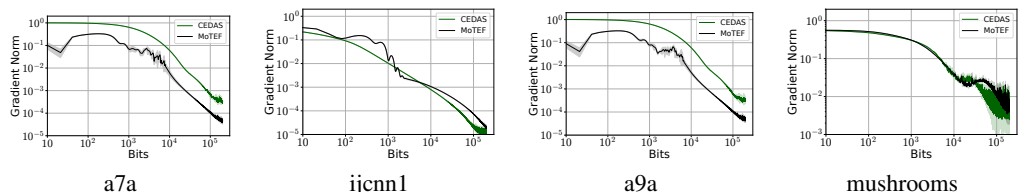

Figure 7: Performance of MoTEF and CEDAS in training logistic regression with non-convex regularization on LibSVM datasets.

## D.6 COMPARISON AGAINST CEDAS

In this section, we consider the comparison against CEDAS algorithm. We demonstrate the performance of MoTEF and CEDAS in the training of logistic regression with non-convex regularization used in Section 4.2. Similarly, we use LibSVM datasets. We tune the parameters $\gamma \in \{10^{-3}, 3 \cdot 10^{-3}, 10^{-2}, 3 \cdot 10^{-2}, 10^{-1}\}, \eta \in \{10^{-4}, 3 \cdot 10^{-4}, 10^{-3}, 3 \cdot 10^{-3}, 10^{-2}, 3 \cdot 10^{-2}, 10^{-1}, 3 \cdot 10^{-1}, 10^0\}, \alpha \in \{10^{-2}, 3 \cdot 10^{-2}, 10^{-1}, 3 \cdot 10^{-1}, 10^0\}$ for CEDAS algorithm, and $\{10^{-3}, 3 \cdot 10^{-3}, 10^{-2}, 3 \cdot 10^{-2}, 10^{-1}\}, \eta \in \{10^{-4}, 3 \cdot 10^{-4}, 10^{-3}, 3 \cdot 10^{-3}, 10^{-2}, 3 \cdot 10^{-2}, 10^{-1}, 3 \cdot 10^{-1}, 10^0\}, \lambda \in \{0.9, 0.8, 0.1\}$ for MoTEF. We use Rand-$K$ compressor for CEDAS and Top-$K$ for MoTEF both with $K = 10$, and mini-batch stochastic gradients with a batch size 16. We compare the performance of algorithms on the ring topology with $n = 10$ and regularization parameter $10^{-1}$ averaging over 3 different random seeds. In Figure 7, we demonstrate the communication performance of CEDAS and MoTEF with the best set of parameters. We observe that the performance of MoTEF and CEDAS is similar on ijcnn1 and mushrooms datasets while MoTEF outperforms CEDAS on a7a and a9a datasets.

