# OpenReview forum: "Towards Faster Decentralized Stochastic Optimization with Communication Compression"
_ICLR.cc/2025/Conference — ICLR 2025 Poster_

### Official Review · Reviewer_TEjT · 2024-10-27

**Soundness:** 3
**Presentation:** 3
**Contribution:** 3
**Rating:** 8
**Confidence:** 4

**Summary:**

The authors propose new algorithms for decentralized nonconvex optimization with heterogeneous functions, communication compression, and calls to stochastic gradients.

**Strengths:**

As far as I know, the state of the art as summarized in the paper and Appendix A is correctly presented. The contributions are important, as nonconvex decentralized optimization is a timely topic with a wide range of applications.

**Weaknesses:**

My main concern is the following. In Table 1, it is stated that convergence is established with respect to E[||nabla.f(x_out)||] for an appropriately chosen x_out, which as its name suggest should be constructed and output by the algorithm. However, the main result, Theorem 1, is established for x_out = bar{x}_t for a random t. The problem is that bar{x} is the average of the local variables x_i, which is not available! So you only prove one half of a valid convergence statement. The second half is that the method achieves a consensus, which in your case corresponds to Omega_3 converging to zero. Reasoning on bar{x} violates the conditions of decentralized optimization, where communication is assumed to be possible only through the network edges, and with compression.

Is x_out = bar{x}_t used in the experiments? In that case this is clearly unfair to the other methods which do not use this unaccessible oracle.

Minor comments on the state of the art:
* The paper about LEAD by Liu et al. "Linear convergent decentralized optimization with compression" has been published at ICLR 2021.
* The title "Randcom: Random communication skipping method for decentralized stochastic optimization" of the paper arXiv:2310.07983 has changed

**Questions:**

Does it follow from Lemma 1 that in the conditions of Theorem 1, Phi^{t+1} <= Phi^t? This would imply that all quantities in (8) remain bounded (ideally, they would be proved to tend to zero).

---

> ### Author Response · Authors · 2024-11-19
> **Rebuttals**
>
> **W1:** This is a good question! We indeed do not have access to the averaged model $\bar{\mathbf{x}}^t$, however, there are several reasons to consider the expected gradient norm at the averaged iterate. First, we highlight that using the average iterate $\bar{\mathbf{x}}^t=$ $\frac{1}{n}\sum_{i=1}^n \mathbf{x}\_{i}^t$ in the convergence metric is a standard technique and was done in many previous works [1-4]. Second, from practical considerations, to obtain an averaged model after the end of MoTEF, we can perform a gossip averaging with compression (e.g., Choco-Gossip [2]) that converges linearly, i.e. adds only logarithmic terms in the convergence rate. Therefore, obtaining the average can be done in practice without hurting the rate. Next, in the revised version of the paper, we strengthened our convergence guarantees of MoTEF. In section B.1.1, we provide detailed derivations that show that the consensus error term $\Omega_3^t = \mathbb{E}\left[\\|\mathbf{X}^t -\bar{\mathbf{x}}^t \mathbf{1}^\top\\|^2\right]$ converges to zero as well. Together with the convergence of $\mathbb{E}[\\|\nabla f(\mathbf{x}_{\rm out})\\|^2]$ we ensure that all local models $\mathbf{x}_i^t$ also converge to stationarity. We refer to Corollary A.6 in [5] for derivations of this claim. We also add the derivations in Section B.1.2.
>
> **W2:** We thank the reviewer for this comment. We adjusted the references accordingly.
>
> **Q1:** We do not obtain the monotonic decrease of the Lyapunov function. It is only possible if the workers use full gradients, i.e. the variance $\sigma^2=0.$ In the stochastic regime, the decrease of the Lyapunov function is done up to the error term which scales with $\sigma.$ Therefore, the current analysis does not automatically imply the convergence of each of the terms in the Lyapunov function. Nonetheless, in the response to W1 we demonstrate that the consensus error term $\Omega_3^t$ converges to zero.
>
> [1] Koloskova, Anastasia and Lin, Tao and Stich, Sebastian U and Jaggi, Martin, Decentralized deep learning with arbitrary communication compression, ICLR, 2020.
>
> [2] Koloskova, Anastasia and Stich, Sebastian and Jaggi, Martin, Decentralized stochastic optimization and gossip algorithms with compressed communication, ICML, 2019.
>
> [3] Zhao, Haoyu and Li, Boyue and Li, Zhize and Richtàrik, Peter and Chi, Yuejie, BEER: Fast $O(1/T)$ Rate for Decentralized Nonconvex Optimization with Communication Compression, NeurIPS, 2022
>
> [4] Huang, Kun and Pu, Shi, Cedas: A compressed decentralized stochastic gradient method with improved convergence, IEEE Transactions on Automatic Control, 2024.
>
> [5] Koloskova, Anastasia and Lin, Tao and Stich, Sebastian U and Jaggi, Martin, Decentralized deep learning with arbitrary communication compression, arXiv preprint arXiv:1907.09356, 2019.

---

> > ### Comment · Reviewer_TEjT · 2024-11-21
> >
> > Thank you for the explanations. Proving that the consensus error tends to zero is crucial, and it is good that you managed to prove it. So, I am increasing my score.

---

### Official Review · Reviewer_DZJC · 2024-10-29

**Soundness:** 4
**Presentation:** 4
**Contribution:** 3
**Rating:** 8
**Confidence:** 4

**Summary:**

Compression has become a key technique in federated learning to address the primary bottleneck of communication efficiency. This paper introduces a new algorithm, MoTEF, designed for decentralized federated learning with communication compression. The distinctive features of MoTEF include integration with model compression, moment tracking and error feedback altogether.

The authors provide a convergence analysis showing that MoTEF achieves some of the best expected results, notably without requiring heterogeneity assumptions. They discuss convergence for general non-convex functions and for functions that satisfy the PL condition (a broader condition than convexity). Additionally, they present a moment-based variance reduction variant of MoTEF.

Theoretical insights into the algorithm are explored through comparisons with existing bounds, as well as through numerical experiments.

**Strengths:**

It is impressive that the authors prove a convergence bound for such a complex algorithm without assuming a specific degree of data heterogeneity. Their other assumptions are also reasonable. Although the bound has a suboptimal dependence on \rho, their experiments demonstrate that the algorithm’s sensitivity to \rho can actually be much lower, offering valuable insights to the community.

The presentation is excellent, with comprehensive discussions that thoroughly compare their results to existing work.

**Weaknesses:**

More explanation to what Algorithm MoTEF actually does can improve the paper. From what is is written, it seems the algotihm just puts togther all the previous tricks into one place.

One minor suggestion: while the authors say "The codes to reproduce our synthetic experiment can be accessed here", the URL is provided at the end of page 9.

**Questions:**

Can you elaborate which of the three trick (GT, moment, error feedback) helps remove the data heterogeneity of the paper?

Can you simplify the bounds in (11)? In particular, there seems to be a tradeoff on \alpha among the second term, third term, fourth term. In other words, can you provide a unifying bounds that incorporates these three terms?

---

> ### Author Response · Authors · 2024-11-19
> **Rebuttals**
>
> **W1:** Even though the mechanisms we used in the design of MoTEF algorithm are known in the literature, combining them is a challenging task and requires careful investigation. For example, we highlight that the order of steps in the algorithm plays a crucial role. In [1] they have similar mechanisms in the algorithm. They use two tracking steps: one is inside the momentum term (line 10 in Alg. 1 [1]) and another one in the update of gradient estimators (line 12 in Alg. 1 [1]). Afterward, the gradient estimators' differences are compressed and aggregated among neighbors. Their algorithm design leads to suboptimal asymptotic convergence. In our algorithm, the gradient tracking step is involved in the update of gradient estimator $V$ only (line 8) involving the averaging from the previous step. Then, new gradient estimators' differences are compressed. Based on the discussion above, we believe that combining contractive compression and stochastic gradients with gradient tracking and Error Feedback in decentralized training is a challenging task as it requires proper algorithm design to overcome all difficulties simultaneously without imposing strong assumptions on the problem.
>
> To improve the writing, we aim to add the following discussion about the algorithm design which is also reported in Section A.1:
>
> Designing an algorithm with strong convergence guarantees without imposing assumptions on the problem or data is complicated. In MoTEF we incorporate three main ingredients to make it converge faster under arbitrary data heterogeneity. In particular, the combination of EF21-type Error Feedback and Gradient Tracking mechanisms is the key factor in getting rid of the influence of data heterogeneity. We emphasize that not using any one of them would lead to restrictions on the data heterogeneity. Indeed, EF21 is known to remove such dependencies in centralized training [2] while the GT mechanism is essential in decentralized learning [3]. Nonetheless, EF21 does not handle the error coming from stochastic gradients and momentum is known to be one of the remedies to it [4].
>
> **W2:** In fact, it was a hyperlink in the text. In the revised version, we changed it to the footnote for visibility. The link on page 9 is to experiments with real data (logistic regression and neural networks).
>
> **Q1:**  This is a great question! First, note that EF21 [2] Error Feedback mechanism removes the data heterogeneity assumption in the **centralized** setting. GT mechanism is needed in the **decentralized** training (without compression) as vanilla decentralized SGD is affected by the data heterogeneity [3]. Therefore, we believe that these two mechanisms are essential to designing an algorithm whose convergence is not affected by data heterogeneity in the decentralized training.
>
> **Q2:** We thank the reviewer for bringing this comment. After carefully checking the derivations, we found several minor typos in the calculations and fixed them. Moreover, we provide a simplified convergence rate in both non-convex and PL regimes. Indeed, one of the middle terms dominates the other two, and therefore, the rate can be simplified. Similarly, we provide the simplified convergence rate of MoTEF-VR in the revised version of the paper.

---

> > ### Author Response · Authors · 2024-11-22
> > **Reminder**
> >
> > Dear reviewer,
> >
> > We would like to remind you that the discussion period ends soon. Therefore, we would like to know if there are any other concerns left unaddressed or should be clarified more. We would be happy to provide any further details to answer them. Thank you!

---

> > ### Comment · Reviewer_DZJC · 2024-11-25
> >
> > I apologize for the delayed response and appreciate the authors' efforts in addressing my earlier questions; most of them seem to have been resolved satisfactorily.
> >
> > However, I have a minor clarification request: could the authors explicitly provide the references corresponding to [1], [2], [3], and [4] mentioned in the comments? I attempted to cross-check them with the references in the paper, but I am uncertain whether they are being referred to in the intended order. Providing these details would be greatly helpful.

---

> ### Author Response · Authors · 2024-11-26
>
> Here are the explicit references. We apologise for the accidental omission in our previous response.
>
> [1] Chung-Yiu Yau and Hoi-To Wai. Docom: Compressed decentralized optimization with near-optimal sample complexity. arXiv preprint arXiv:2202.00255,
> 2022.
>
> [2] Richtàrik, Peter and Sokolov, Igor and Fatkhullin, Ilyas, EF21: A new, simpler, theoretically better, and practically faster error feedback, NeurIPS, 2021.
>
> [3] Koloskova, Anastasia and Loizou, Nicolas and Boreiri, Sadra and Jaggi, Martin and Stich, Sebastian, A unified theory of decentralized sgd with changing topology and local updates, ICLR, 2020.
>
> [4] Fatkhullin, Ilyas and Tyurin, Alexander and Richtàrik, Peter, Momentum provably improves error feedback!, NeurIPS, 2024.

---

### Official Review · Reviewer_SzmB · 2024-10-30

**Soundness:** 2
**Presentation:** 3
**Contribution:** 2
**Rating:** 6
**Confidence:** 3

**Summary:**

This paper proposes MoTEF which achieves faster asymptotic convergence rate on decentralized optimization with communication compression, without using strong assumptions such as bounded gradient, bounded heterogeneity or unbiased compression. A variance-reduction version called MoTEF-VR is also introduced. Ablation studies show that MoTEF enjoys linear speed-up and is robust to network topology. Numerical experiments show that MoTEF performs better than Choco-SGD and BEER.

**Strengths:**

1. This work achieves the fastest asymptotic convergence rates with weakest assumptions.
2. The presentation is neat and clear.

**Weaknesses:**

1. The improvement on theoretical convergence result is not significant. Compared to CEDAS, it seems that the only improvement is removing the need for an additional unbiased compressor. To better illustrate this improvement, it is expected to validate whether using contractive compressors are more efficient than using unbiased ones. Otherwise, maybe the authors can compare the full convergence complexity (instead of the asymptotic one only) to address the theoretical improvement.
2. The numerical experiments are not persuasive enough. The compared baselines are Choco-SGD and BEER, which are in 2022 or earlier, and their convergence rate is clearly worse than SOTA as illustrated in Table 1. In contrast, CEDAS that seems closer to SOTA convergence rate is not compared. Maybe the authors can make the experimental results more solid by adding more baselines like CEDAS and DeepSqueeze.

**Questions:**

1. Can the authors better illustrate the advantage of MoTEF against CEDAS both theoretically and empirically? For example, in what sense using contractive compression is better than unbiased compression, and whether MoTEF can perform better than CEDAS?
2. The result for CNN seems missing. Please make sure to include both the results and the implementation details.

---

> ### Author Response · Authors · 2024-11-19
> **Rebuttals**
>
> **W1:** First, MoTEF provably converges with contractive compressors (e.g., Top-K) while CEDAS converges only with unbiased compressors (e.g., Random-K). The class of contractive compression operators is more general and contains operators such as Top-K that do not have unbiased property. Many earlier works demonstrated that contractive compressors are superior in practice [3-5], known to be near-optimal in theory [1], achieve smaller variance both theoretically and empirically than their unbiased cousins [2]. Moreover, combining unbiased compressor with contractive one improves the practical performance [6]. Second, MoTEF uses momentum mechanism which is known to accelerate the performance both theoretically [7] and practically [8-9] in training deep models.
>
> **W2:** We provide the empirical comparison of MoTEF and CEDAS algorithms on the logistic regression with non-convex regularization in section D.6. We demonstrate that MoTEF achieves smaller gradient norm in most of the cases that showcase its practical superiority as well.
>
> **Q1:** we refer to the responses to **W1** and **W2**
>
> **Q2:** We thank the reviewer for pointing out the typo. In the revised version of the paper, Figure 6 contains the empirical results in training CNN model on MNIST dataset and Section D.5 describes the training details and results.
>
> [1] Albasyoni, Alyazeed and Safaryan, Mher and Condat, Laurent and Richtárik, Peter, Optimal gradient compression for distributed and federated learning, arXiv preprint arXiv:2010.03246, 2020
>
> [2] Beznosikov, Aleksandr and Horváth, Samuel and Richtárik, Peter and Safaryan, Mher, On biased compression for distributed learning Journal of Machine Learning Research, 2023
>
> [3] Lin, Yujun and Han, Song and Mao, Huizi and Wang, Yu and Dally, William J, Deep gradient compression: Reducing the communication bandwidth for distributed training, arXiv preprint arXiv:1712.01887, 2017.
>
> [4] Haobo Sun and Yingxia Shao and Jiawei Jiang and Bin Cui and Kai Lei and Yu Xu and Jiang Wang, Sparse Gradient Compression for Distributed SGD, International Conference on Database Systems for Advanced Applicationsl 2019.
>
> [5] Vogels, Thijs and Karimireddy, Sai Praneeth and Jaggi, Martin, PowerSGD: Practical low-rank gradient compression for distributed optimization, NeurIPS, 2019.
>
> [6] Horváth, Samuel and Richtárik, Peter, A better alternative to error feedback for communication-efficient distributed learning, ICLR, 2021.
>
> [7] Cutkosky, Ashok and Mehta, Harsh, Momentum improves normalized sgd, ICML, 2020.
>
> [8] Choi, Dami and Shallue, Christopher J and Nado, Zachary and Lee, Jaehoon and Maddison, Chris J and Dahl, George E, On empirical comparisons of optimizers for deep learning, arXiv preprint arXiv:1910.05446, 2019.
>
> [9] Fu, Jingwen and Wang, Bohan and Zhang, Huishuai and Zhang, Zhizheng and Chen, Wei and Zheng, Nanning, When and why momentum accelerates sgd: An empirical study, arXiv preprint arXiv:2306.09000, 2023.

---

> > ### Author Response · Authors · 2024-11-22
> > **Reminder**
> >
> > Dear reviewer,
> >
> > We would like to remind you that the discussion period ends soon. Therefore, we would like to know if there are any other concerns left unaddressed or should be clarified more. We would be happy to provide any further details to answer them. Thank you!

---

> > ### Comment · Reviewer_SzmB · 2024-11-23
> > **Further concerns**
> >
> > Thanks for the response and the additional experiments on CEDAS. There remains several concerns to be addressed.
> >
> > 1. While the theoretical improvement of MoTEF that it ensures convergence on contractive compressors while CEDAS only ensures convergence on unbiased compressors is clear, the reason why contractive compression is better than unbiased compression is not quite clear. Although [1] presents the near-optimal results of contractive compressors, I consider unbiased compressors can also reach this limit. Specifically, I believe the limit in [1] is reachable by rand-K compressors, which have unbiased cousins.
> >
> > 2. I have some suggestions in the additional experiments. First, it seems that in three out of four experiments, CEDAS cannot converge precisely, which seems contradict to its convergence proofs. It's possible that hyperparameters are not chosen properly. Second, it is suggested to compare the communicated bits rather than the number of iterations between two algorithms, which is more consistent in the paper, and makes fairer comparisons. It appears that CEDAS only communicates once per iteration and MoTEF communicates twice. If so, comparing them in the number of iterations is unfair.

---

> ### Author Response · Authors · 2024-11-23
> **Response to the reviewer**
>
> 1. Thank you for your response. We would like to address the remaining concerns.
> The lower bound  (also called uncertainty principle for communication compression) presented in [1] shows that a compression scheme with a distortion $\alpha$ and $b$ encoding bits (in the worst case) satisfies
> $$b \approx \frac{d}{2}\log\frac{1}{\alpha},$$
> where we ignore $\mathcal{O}(\log d)$ terms. [1] shows that there is an example of biased compressor which matches the lower bound. For the unbiased $Q$ compressor with parameter $\omega > 0$ (for example, for Rand-k $\omega = \frac{d}{k}-1$) we have that $\frac{1}{1+\omega}Q$ is the biased compressor with parameter $\alpha=\frac{\omega}{1+\omega}$. Plugging this number in the lower bound we get that for the unbiased compressor it matches the lower bound if
> $$b \approx \frac{d}{2}\log(1+1/\omega).$$
> Particularly, in the case of Rand-k compressor we have $\omega=\frac{d}{k}-1$, and therefore we should have
> $$b \approx \frac{d}{2}\log(d/(d-k)).$$
> However, for Rand-k compressor we have $b = 32k \log d$ which is larger than $\frac{d}{2}\log(d/(d-k))$ if $k$ is sufficiently small than $d$. Therefore, Rand-k compressor does not satisfy the lower bound. This is also illustrated in [10], Figure 1 where the authors demonstrate that biased compressors are closer to the lower bound than unbiased ones.
> Moreover, we also refer to Lemma 20 in [2] where the authors analyze the difference in the compression error of Top-k and Rand-k compressors, In particular, they demonstrate that if the entries of the input follow the standard exponential distribution then Top-K compressor's error is much smaller than that of Rand-k. In Figure 1 they also demonstrate that Top-k compressor in average requires less bits to encode one entry than Rand-k with the same normalized variance. Finally, Figure 2 in [2] showcases that Top-k compressor saves "more information" about the input than Rand-k for practical gradient distributions.
>
>     These observations demonstrate the superiority of biased compressors both theoretically and practically.
>
>     [1] Albasyoni, Alyazeed and Safaryan, Mher and Condat, Laurent and Richtárik, Peter, Optimal gradient compression for distributed and federated learning, arXiv preprint arXiv:2010.03246, 2020
>
>     [2] Beznosikov, Aleksandr and Horváth, Samuel and Richtárik, Peter and Safaryan, Mher, On biased compression for distributed learning, Journal of Machine Learning Research, 2023
>
>     [10] Safaryan, Mher and Shulgin, Egor and Richtarik, Peter, Uncertainty principle for communication compression in distributed and federated learning and the search for an optimal compressor, A Journal of the IMA, 2022.
>
> 2. We will provide further experiments soon with a larger grid search in terms of gradient norm vs transmitted bits.

---

> ### Author Response · Authors · 2024-11-24
> **Response to the reviewer**
>
> 2. Additional experimental results.
>    - In Figure 7, we compare MoTEF and CEDAS in terms of gradient norm vs. bits. We increased the parameter set for the step sizes; please see the description in Section D.6. We took into account that the communication of one step of MoTEF is two times larger since workers exchange messages twice per iteration. We observe that MoTEF either matches CEDAS or outperforms it in terms of communication complexity.
>
>     - We would like to emphasize that we demonstrate the best performance that is achievable by setting the parameters from the corresponding set. From the empirical observations, we observe that CEDAS requires smaller step-size parameters $\gamma$ and $\eta$ to achieve the same gradient norm as MoTEF. However, choosing too small step-sizes $\gamma$ and $\eta$ leads to a significantly slower convergence speed of CEDAS.

---

> > ### Comment · Reviewer_SzmB · 2024-11-24
> > **Thank you for the rebuttal**
> >
> > Thanks for the detailed response.
> > 1. For point 1, I consider the bits for Randk should be $32k+\log_2\binom{d}{k}$ with $32k$ bits representing the $k$ entries and $\log_2\binom{d}{k}$ bits representing the selected $k$ indicies. Anyway, the order $\Theta(k\log d)$ remains the same, which is larger than the lower bound by a factor of $\Theta(\log d)$. The authors are right that RandK can not reach the limit while I have previously overlooked the logarithm term habitually.
> > 2. Thank you for presenting fairer comparisons with CEDAS, where MoTEF performs slightly better than CEDAS in two of them and performs similarly in the others.
> >
> > I have no further questions, and I'll raise my score to 6. Good luck!

---

### Official Review · Reviewer_DwbG · 2024-11-01

**Soundness:** 3
**Presentation:** 3
**Contribution:** 3
**Rating:** 6
**Confidence:** 4

**Summary:**

This paper studies decentralized stochastic optimization with communication compression. It introduces the momentum tracking technique with error feedback, and achieves the first linear speedup convergence rate under the standard assumptions. Numerical experiments are conducted to validate the theoretical findings.

**Strengths:**

1. It combines momentum tracking and error feedback to attain an effective compressed decentralized algorithm.

2. It achieves the first linear speedup convergence rate for decentralized algorithms with contractive compressors.

**Weaknesses:**

1. The novelty seems a little bit limited. The main idea and analysis techniques seems to be a direct extension of the centralized algorithm EControl (Gao et.al., 2024) to decentralized settings.

2. The insight behind the proposed algorithm is not well clarified. Why does the combination of the momentum tracking and error feedback result in the linear speedup rate? It is encouraged to discuss how the algorithms are developed and highlight the insight.

3. The dependence on the network topology, as the authors have discussed, is much worse than decentralized algorithms without compression.

**Questions:**

1. Please highlight the challenges in analysis and algorithmic developments compared to the EControl algorithm (Gao et.al., 2024).

2. Please have an in-depth discussion on how the algorithm is developed. Why does the combination of the momentum tracking and error feedback result in the linear speedup rate?

3. If there is no communication compression and error feedback, does your algoithm reduce to the pure momentum tracking algorithm? How does this momentum tracking algorithm compare with the well-known gradient tracking algorithm in convergence rate?

4. In your Theorem 1, if the network is fully connected, i.e., rho=1, how does your algorithm compare with state-of-the-art centralized compressed algorithm such as EControl, Error-feedback with momentum, and NEOLITHC?

5. The numerical studies are a little bit trivial. In your MLP task, what dataset did you use? Can you evalaute your algorithm over more realistic tasks, such as ResNet on Cifar10?

---

> ### Author Response · Authors · 2024-11-19
> **Rebuttal**
>
> **W1:** We would like to clarify the main differences between MoTEF and EControl. First, MoTEF incorporates EF21/CHOCO-style Error Feedback while EControl uses a more classical Error Compensation mechanism [9]. Having different error mechanisms leads to significantly different analysis techniques (e.g., analysis via virtual iteration vs. a more direct proof, and the design of the Lyapunov function: their Lyapunov function has only 3 terms while ours has 6, i.e. it requires much more involved and technical analysis). Next, EControl archives linear speedup by properly balancing the error term $e^t$, and the error carried from the gradient estimator. In our work, this is done by incorporating momentum which reduces the variances. Finally, in our work we provide the convergence guarantees under PL condition and the convergence of MoTEF-VR that was not done in EControl paper.
>
> **W2:** Designing an algorithm with strong convergence guarantees without imposing assumptions on the problem or data is complicated. In MoTEF we incorporate three main ingredients to make it converge faster under arbitrary data heterogeneity. In particular, the combination of EF21-type Error Feedback and Gradient Tracking mechanisms is the key factor in getting rid of the influence of data heterogeneity. We emphasize that not using one of them would lead to restrictions on the data heterogeneity. Indeed, EF21 is known to remove such dependencies in centralized training [1] while the GT mechanism is essential in decentralized learning [2]. Nonetheless, EF21 does not handle the error coming from stochastic gradients and momentum is known to be one of the remedies for it [3]. Without the momentum term, EF21 is known to converge only with large enough batches in the centralized stochastic regime [3]. Moreover, we emphasize that earlier work [10] incorporates similar ideas but fails to achieve optimal asymptotic rate due to the incorrect order of mechanisms in their algorithm.
>
> With appropriately chosen parameters, the momentum technique reduces the variance (between the momentum and the gradient) as the algorithm proceeds. On a more technical level, this variance-reduction property enables us to do a descent analysis on $\hat G^t$, the variance between the **averaged** momentum and the local gradients. Being able to reason with the **averaged** momentum is crucial for achieving the linear speedup because it enables us to analyze the averaged noises (whose variance is reduced linearly in $n$), instead of the individual noises, at each iteration.
>
> **Q1:** we refer to the response to **W1**
>
> **Q2:** we refer to the response to **W2**
>
> **Q3:** This is a great question. Indeed, if we set the compression operator to be identity, momentum parameter $\lambda = 1- \beta$, and slightly modify the variables in MoTEF, then the momentum tracking algorithm from [1] is almost identical to our method. The main difference comes from the fact we also use a mixing step with stepsize $\gamma$ which is needed because of the use of compression (see [5], for instance). [4] also obtain optimal asymptotic rate as we do, but their deterministic rate has slightly better dependency on the spectral gap $\rho:$ $\rho^{5/2}$ instead of $\rho^3$ in our work. However, we highlight that combining momentum tracking and compression is a challenging task as naive use of contractive compressors might lead to divergence [6]. Therefore, the Error Feedback mechanism is needed to tackle this issue. This translates into more involved and technical proofs. The convergence rate of GT has $\mathcal{O}(\frac{\sigma^2}{n\varepsilon^4} + \frac{\sigma}{(\rho^{3/2} + \rho\sqrt{n})\varepsilon^{3}} + \frac{1}{\rho^2\varepsilon})$ [7]. We observe that both MoTEF and GT achieve optimal asymptotic rate, but GT has slightly better dependency on the spectral gap in the deterministic regime: $\rho^2$ instead of $\rho^3.$
>
> **Q4:**  If we set $\rho=1$ in the convergence rate of MoTEF, we obtain the asymptotic $\frac{LF^0\sigma^2}{n\varepsilon^4}$ and deterministic $\frac{LF^0}{\alpha\varepsilon^2}$ rates that match those of EControl, EF21-SGDM, and Neolithic. The difference between algorithms is in the middle term(s) in the rate. Neolithic does not have this term since the analysis is performed under a more restricted assumption of the bounded gradient dissimilarity and performing an impractical multi-stage compression mechanism (i.e., several communication rounds per iteration). The middle term in the convergence rate of EControl has a worse dependency on $\alpha:$ $\alpha^2$ instead of $\alpha$ in our work. The convergence rate of EF21-SGDM has two middle terms. The worst of them scales scales with $\alpha^{1/2}$ while it is $\alpha$ in the convergence of MoTEF.

---

> ### Author Response · Authors · 2024-11-19
> **Rebuttal (part 2)**
>
> **Q5:** In the experiments with MLP model, we use MNIST dataset. Moreover, we point out that we provide the experiments with CNN model on MNIST dataset in the appendix. The main contribution of our work is a novel MoTEF and MoTEF-VR algorithms with their convergence analysis. We provide convergence guarantees in general non-convex and PL regimes. We support our theoretical findings in training logistic regression with non-convex regularization as well as training of MLP and CNN (in the appendix) models. In all the cases, the empirical results demonstrate the superiority of MoTEF algorithm. Moreover, we provide the experiments that showcase the robustness of MoTEF algorithm to the changes of the network topology. For a more comprehensive experimental study with more complex models and real-life decentralized training hardware, we defer them to a more experiment-focused work in the future due to resource constraints.
>
> [1] Richtàrik et al., EF21: A new, simpler, theoretically better, and practically faster error feedback, NeurIPS, 2021.
>
> [2] Koloskova et al., A unified theory of decentralized sgd with changing topology and local updates, ICLR, 2020.
>
> [3] Fatkhullin et al., Momentum provably improves error feedback!, NeurIPS, 2024.
>
>
> [4] Y. Takezawa et al., Momentum tracking: momentum acceleration for decentralized deep learning on heterogeneous data, TMLR, 2023.
>
> [5] Koloskova et al., Decentralized stochastic optimization and gossip algorithms with compressed communication, ICML 2019, 2019.
>
> [6] Beznosikov et al., On biased compression for distributed learning, JMLR, 2023.
>
> [7] Koloskova et al., An improved analysis of gradient tracking for decentralized machine learning, NeurIPS, 2021.
>
> [8] Di et al., Double Stochasticity Gazes Faster: Snap-Shot Decentralized Stochastic Gradient Tracking Methods, ICML, 2022.
>
> [9] Seide et al., 1-bit stochastic gradient descent and its application to data-parallel distributed training of speech DNNs, Interspeech, 2014.
>
> [10] Yau \& Wai. Docom: Compressed decentral-
> ized optimization with near-optimal sample complexity. arXiv preprint arXiv:2202.00255, 2022

---

> > ### Author Response · Authors · 2024-11-22
> > **Reminder**
> >
> > Dear reviewer,
> >
> > We would like to remind you that the discussion period ends soon. Therefore, we would like to know if there are any other concerns left unaddressed or should be clarified more. We would be happy to provide any further details to answer them. Thank you!

---

> > ### Comment · Reviewer_DwbG · 2024-11-26
> > **Thanks for the rebuttal**
> >
> > Thank you for your thoughtful response. While some of your points have addressed my concerns, the paper's worse dependency on network topology remains an issue. I believe my current rating accurately reflects the paper's value.

---

### Official Review · Reviewer_rAXz · 2024-11-04

**Soundness:** 3
**Presentation:** 3
**Contribution:** 3
**Rating:** 5
**Confidence:** 4

**Summary:**

This paper proposes a novel approach MoTEF to achieve an asymptotic rate matching that of distributed SGD under arbitrary data heterogeneity by adding momentum tracking and error feedback technique, solving a theoretical obstacle in decentralized optimization with compression. This paper conducts numerical experiments to illustrate the effectiveness of MoTEF.

**Strengths:**

1. MoTEF achieves the convergence rate matching distributed SGD without strong assumptions, such as bounded gradient or global heterogeneity bound. It is an important improvement in distributed optimization with compression.
2. MoTEF supports arbitrary contractive compressors (variance-bounded estimate) without unbiasedness.
3. Extension MoTEF to the stochastic setting can achieve an improved rate with variance reduction.
4. This paper proposes theoretical analysis under the PL condition.

**Weaknesses:**

1. The comparison needs to be more clarified and detailed. Especially, the total communication complexity is important in optimization with compression. Most compression algorithms can only reduce the communication overhead of single-step iteration, but cannot reduce the total communication overhead required for convergence. It is necessary to discuss it in detail.
2. Though the numerical experiments are enough to illustrate the effectiveness of MoTEF, more evidences in practical problems are necessary. For example, a lightweight training on transformers instead of only MLP.

**Questions:**

1. Though the proof is clear enough, I am interested in the insight of the construction of the  Lyapunov function. Adding an overview of the technique before the theoretical results is better.
2. It is a valuable study. If the author explains my concerns, I would like to improve my score.

---

> ### Author Response · Authors · 2024-11-19
> **Rebuttal**
>
> **W1:** We thank the reviewer for this valuable comment. In this work, we consider the general contractive compression with the parameter $\alpha$ which quantifies the compression quality instead of the compression ratio directly. This definition covers many popular compressors in practice, for which we can discuss the total communication complexity. Below we provide a comparison of total communication complexity based on Top-$K$ compressor which is a contractive compressor with $\alpha =K/d$. In each round of communication, a client sends data of size proportional to $K$ (up to $\log(K)$ terms) to its neighbor, instead of the dimension $d$. Therefore, the total communication complexity of the algorithm with Top-$K$ compressor is proportional to $K \times \text{ number of iterations }$,
> while for the non-compressed methods, the total communication complexity is $d \times \text{ number of iterations }$. Now plugging our complexity bound in the paper, we have that the total communication complexity of MoTEF with Top-$K$ becomes:
> $$
>     \mathcal{O}\left(\frac{\sigma^2 K}{n\varepsilon^4} + \frac{\sigma d}{\rho^{5/2}\varepsilon^3} + \frac{d}{\rho^3\varepsilon^2}\right),
> $$
> while the uncompressed decentralized SGD with gradient tracking has:
> $$
>     \mathcal{O}\left(\frac{\sigma^2 d}{n\varepsilon^4} +  \frac{d}{\rho^2\varepsilon^2}\right).
> $$
> In the deterministic regime, i.e. when $\sigma^2=0$, our method obtains a $\frac{d}{\rho^3\varepsilon^2}$ rate, which is slightly worse than the rate of the uncompressed method by a factor of $1/\rho$. This is however negligible when the graph is moderately well-connected. More importantly, in the noisy regime which is more typical in the modern machine learning setting, the asymptotically dominant term for MoTEF is $\mathcal{O}(\frac{\sigma^2 K}{n\varepsilon^4})$ which has a $K/d$ factor of improvement over the uncompressed method, $\mathcal{O}(\frac{\sigma^2 d}{n\varepsilon^4})$.
>
> **W2:** The main contribution of our work is a novel MoTEF and MoTEF-VR algorithms with their convergence analysis. We provide convergence guarantees in general non-convex and PL regimes. We support our theoretical findings in training logistic regression with non-convex regularization as well as training of MLP and CNN (in the appendix) models. In all the cases, the empirical results demonstrate the superiority of MoTEF algorithm. Moreover, we provide the experiments that showcase the robustness of MoTEF algorithm to the changes of the network topology. For a more comprehensive experimental study with more complex models and real-life decentralized training hardware, we defer them to a more experiment-focused work in the future due to resource constraints.
>
> **Q1:** We appreciate the interest of the reviewer in the design of the Lyapunov function. First, the term $F^t$ typically appears in the Lyapunov-type analysis in the non-convex regime. Second, two terms $\Omega_1^t$ and $\Omega_2^t$ are consensus errors that are present in the Lyapunov function as we consider the decentralized training, and local models and gradient estimators eventually should be close to each other. Next, due to presence of the compression there are additional terms $\Omega_3^t$ and $\Omega_4^t$ to control the compression error. Finally, the terms $\hat{G}^t$ and $\tilde{G}^t$ are there to control the difference between full gradient $\nabla F(X^t)$ and momentum $M^t$, i.e. to showcase that $M^t$ is a good enough approximation of the true gradient. We emphasize that even though the convergence can be shown without using $\hat{G}^t$ but this term is crucial for proving the linear speedup with $n$ in the asymptotic term $\frac{\sigma^2}{n\varepsilon^4}$. Therefore, it is important to have both terms in the Lyapunov function. The coefficient next to each term are chosen to balance the descent rates of all the terms.

---

> > ### Author Response · Authors · 2024-11-22
> > **Reminder**
> >
> > Dear reviewer,
> >
> > We would like to remind you that the discussion period ends soon. Therefore, we would like to know if there are any other concerns left unaddressed or should be clarified more. We would be happy to provide any further details to answer them. Thank you!

---

> > > ### Comment · Reviewer_rAXz · 2024-11-26
> > >
> > > I apologize for the delayed response and appreciate your clarification to my concerns. The explanation of the communication complexity and the construction of the Lyapunov function is clear enough. However, I think in optimization area (rather than "learning theory"), it is necessary to evaluate the method in real-life practical tasks, even if the focus is on theoretical analysis. The application could be lightweight like [1], but only "toy model" is not enough.
> > >
> > > [1] Liu H, Li Z, Hall D, et al. Sophia: A scalable stochastic second-order optimizer for language model pre-training[J]. arXiv preprint arXiv:2305.14342, 2023.

---

> > > > ### Author Response · Authors · 2024-12-02
> > > >
> > > > We understand the reviewer’s interests in our algorithm’s practical performances in larger scale experiments. While we are currently constrained by computational resources, we are actively working on extending our experiments to larger scales, although these efforts may not be completed within the rebuttal period.
> > > >
> > > > We thank the reviewer for appreciating the importance of our work for its advancements in the theory of decentralized optimization with communication compression. We believe these theoretical insights represent a meaningful contribution to the field, independent of the scale of experiments, and we are committed to exploring their practical implications further in future work.

---

### Author Response · Authors · 2024-11-19
**General response to all reviewers**

We thank the reviewer very much for their dedication to the review process and for taking the time to carefully study our manuscript. We provide detailed responses to all raised concerns. Moreover, we made several changes to the paper and highlighted them in blue color. In particular, $(i)$ we simplified the convergence rate of MoTEF and MoTEF-VR algorithms by removing non-dominant terms; see (10, 11, 13); $(ii)$ we fixed the section D.5 that contains experiments in the training CNN model on MNIST dataset. Now we provide the correct plots of convergence; $(iii)$ we added a comparison against CEDAS in section D.6; $(iv)$ we tightened the analysis of MoTEF algorithm. In section B.1.1 we provide a more accurate analysis that demonstrates that the consensus error $\Omega_3^t = \mathbb{E}[\\|\mathbf{X}^t - \bar{\mathbf{x}}^t\mathbf{1}^\top\\|\_{\mathrm{F}}^2]$ converges to zero. This implies that the local models $\\{\mathbf{x}\_i^t\\}_{i=1}^n$ also converge; see the derivations in Section B.1.2.

---

### Author Response · Authors · 2024-12-04
**Summary of the discussion period**

Dear reviewers,

As the discussion period comes to an end, we would like to summarize it now.

1) All reviewers highlighted the importance of our theoretical contribution. In particular, convergence under arbitrary data heterogeneity and arbitrary contractive compression (reviewers rAXz, DZJC, SzmB).
2) We provided a detailed comparison of biased and unbiased compressors in the response to reviewer SzmB where we highlight that biased compression schemes are superior in practice, achieve smaller variance both theoretically and practically, and are known to match the lower bound with distortion $\alpha$ and $b$ encoding bits. This is particularly important in the comparison with decentralized algorithms that rely on unbiased compression (e.g., CEDAS). To support these claims empirically, we added the comparison against CEDAS algorithm on non-convex logistic regression.
3) We improved the convergence of MoTEF by demonstrating that the consensus error $\Omega_3^t = \mathbb{E}[\\|\mathbf{X}^t - \bar{\mathbf{x}}^t \mathbf{1}^\top \\|^2_{\mathrm{F}}]$. This implies that not only the average models converges, but each local models as well.
4) We demonstrated that MoTEF matches the asymptotic and deterministic rates of state-of-the-art centralized algorithms if we set the spectral gap $\rho=1$. Moreover, we showed that MoTEF improves the communication complexity in the asymptotic regime.
5) We acknowledge the importance of providing the comparison of MoTEF algorithm against other baselines in training larger models. We work on extending the empirical comparison against other baselines.

Best regards,

Authors

---

### Meta-Review · Area_Chair_QDAc · 2024-12-10

**Metareview:**

The paper proposed a decentralized stochastic optimization algorithm with communication compression with momentum tracking and error feedback. While the algorithmic ingredients are not new per se, the assembly and analysis that leads to a state-of-the-art convergence rate for an important problem is worth acceptance.

**Additional Comments On Reviewer Discussion:**

While some reviewer has suggested including larger scale experiments, the AC finds the current experiment adequate for an optimization-focused paper. Improving the network dependency, as raised by some reviewer, will be an area of interest for future research.

---

### Decision · Program_Chairs · 2025-01-22

Accept (Poster)